# Fructooligosaccharides benefits on glucose homeostasis upon high-fat diet feeding require type 2 conventional dendritic cells

Adélaïde Gélineau [1], Geneviève Marcelin[2], Melissa Ouhachi[1], Sébastien Dussaud [1], Lise Voland[2], Raoul Manuel[1], Ines Baba[1], Christine Rouault[2], Laurent Yvan-Charvet [3], Karine Clément [2,4], Roxane Tussiwand [5], Thierry Huby[1] & Emmanuel L. Gautier [1] ✉

Diet composition impacts metabolic health and is now recognized to shape the immune system, especially in the intestinal tract. Nutritional imbalance and increased caloric intake are induced by high-fat diet (HFD) in which lipids are enriched at the expense of dietary fibers. Such nutritional challenge alters glucose homeostasis as well as intestinal immunity. Here, we observed that short-term HFD induced dysbiosis, glucose intolerance and decreased intestinal RORγt⁺ CD4 T cells, including peripherally-induced Tregs and IL17-producing (Th17) T cells. However, supplementation of HFD-fed male mice with the fermentable dietary fiber fructooligosaccharides (FOS) was sufficient to maintain RORγt⁺ CD4 T cell subsets and microbial species known to induce them, alongside having a beneficial impact on glucose tolerance. FOS-mediated normalization of Th17 cells and amelioration of glucose handling required the cDC2 dendritic cell subset in HFD-fed animals, while IL-17 neutralization limited FOS impact on glucose tolerance. Overall, we uncover a pivotal role of cDC2 in the control of the immune and metabolic effects of FOS in the context of HFD feeding.

Westernized dietary patterns have been associated to the outbreak and sharp increase in noncommunicable diseases (NCDs) over the past decades[1,2]. Among NCDs, chronic metabolic disorders are naturally influenced by feeding habits. Indeed, while a healthy diet positively impacts host metabolism, changes in the quantity and quality of the dietary intake can favor the onset and persistence of metabolic dysfunctions[3]. Besides metabolism, the diet is now recognized to impact other organismal functions such as immunity, and this especially holds in the intestinal tract[4].

Earlier studies revealed that diets low in vitamins A or aryl hydrocarbon receptor (AhR) ligands alter intestinal lymphocytes homeostasis[5]. Lately, fermentable dietary fibers have received considerable attention with regards to their widespread impact on immune cells and intestinal homeostasis[6], adding to their well-known ability to prevent metabolic disorders[7]. Importantly, fibers are a major source of energy for the intestinal flora, and thus largely contribute to gut microbiota ecology[8]. Thus, the tightly regulated cross-talk between the gut microbiota and the host immune system is influ-

[1]Sorbonne Université, Institut National de la Santé et de la Recherche Médicale, Inserm, Research Unit on Cardiovascular and Metabolic Diseases, Hôpital de la Pitié-Salpêtrière, Paris, France. [2]Sorbonne Université, Institut National de la Santé et de la Recherche Médicale, Inserm, Nutrition and Obesities: Systemic approaches research group, NutriOmics, Paris, France. [3]Institut National de la Santé et de la Recherche Médicale, Inserm, Université Côte d'Azur, Centre Méditerranéen de Médecine Moléculaire (C3M), Atip-Avenir, Fédération Hospitalo-Universitaire (FHU) Oncoage, Nice, France. [4]Assistance Publique-Hôpitaux de Paris, Hôpital de la Pitié-Salpêtrière, service de Nutrition, Paris, France. [5]National Institute of Dental and Craniofacial Research, National Institutes of Health, Bethesda, MD, USA. ✉e-mail: emmanuel-laurent.gautier@inserm.fr

enced by the dietary intake, which in turn shapes the metabolites, the microbiome community, and the intestinal tract immune subset composition[9].

Unbalanced diets such as the high-fat diet (HFD) are usually low in fibers, which promotes the development of metabolic disorders[10,11]. HFD intake induces alterations across several organs, including the intestine that contributes to systemic metabolic homeostasis by controlling glucose and lipid metabolism[12–14]. Further, HFD-induced alteration of intestinal immune cells homeostasis concurs to precipitate metabolic dysfunctions[15,16]. Notably, HFD feeding was shown to alter intestinal lymphocytes by hampering the maintenance of RORγt-expressing CD4 T cells[17]. Nonetheless, the mechanisms and immune cell subsets that translate dietary cues into intestinal CD4 T cells homeostasis under HFD feeding remain unexplained.

In the intestine, RORγt⁺ CD4 T cells comprise two main populations. It includes regulatory T cells (Tregs) known to be induced peripherally by the microbiota (pTregs)[18]. These cells are involved in tolerance to the microbiota to avoid type 2 immunity[19]. The second population consists of IL-17-expressing T helper (Th17) cells known to be induced in response to commensals, especially segmented filamentous bacteria (SFB)[20]. At the steady state, Th17 cells display a non-inflammatory phenotype and are involved in the regulation of the intestinal barrier[21–23].

Here, we report that short-term HFD (4 weeks) is sufficient to deregulate the homeostasis of Th17 cells and RORγt⁺ regulatory T cells in the intestinal tract as both subsets decreased. HFD feeding was accompanied by dysbiosis, including a reduction in microbial species known to support RORγt⁺ pTregs and Th17 cells development. Importantly, supplementation of HFD-fed animals with the fermentable dietary fiber fructooligosaccharides (FOS) prevents the loss of several of these microbial species and is sufficient to preserve both RORγt⁺ pTregs and Th17 cells in HFD-fed animals. We also reveal that FOS requires type 2 conventional CD11b⁺ dendritic cells (cDC2) to maintain intestinal Th17 cells, but not RORγt⁺ pTregs, and improve glucose homeostasis in HFD-fed animals. Finally, the impact of FOS intake on glucose tolerance was limited following IL-17 neutralization. Overall, we were able to functionally link FOS intake to the cDC2 compartment and show that this dendritic cell subset is critical to the control of the immune and metabolic effects of FOS.

## Results

### HFD feeding decreases RORγt⁺ pTregs and Th17 cells in the small intestine and colon

Prolonged high-fat diet (HFD) feeding was previously shown to alter the intestinal immune system[15,16]. As the gut rapidly adapts to nutritional changes, we asked whether intestinal immune cells are impacted during the first weeks of HFD. Thus, mice were administered either HFD or regular chow diet for 4 weeks. Already at this timepoint, HFD increased body weight (Fig. 1A), weight gain (Fig. 1B), epididymal fat mass (Fig. 1C), and whole-body fat mass (Fig. 1D), but not lean mass (Fig. 1D). Furthermore, after 4 weeks of HFD, animals were glucose intolerant as circulating glucose clearance was inefficient after glucose intake during a glucose tolerance test (Fig. 1E). They also displayed an elevated HOMA-IR index (Fig. 1F), indicative of impaired glucose control by insulin. Higher HOMA-IR reflected both elevated fasting blood glucose (Fig. 1G) and insulin levels (Fig. 1H). Altogether, 4 weeks of HFD feeding altered systemic glucose homeostasis. Morphological changes in the intestinal tract were also evident in HFD-fed animals, including decreased colon weight (Fig. 1I) and length (Fig. 1J), which are commonly associated with altered colonic homeostasis. In addition, and as previously reported[10], HFD feeding resulted in decreased cecum weight (Fig. 1K).

We next asked how intestinal lymphocytes were adapting to HFD and focused on RORγt⁺ CD4 T cells, known to be altered under these conditions[17]. RORγt⁺ CD4 T cells comprise IL-17-producing effector

T cells[24] and a population of regulatory T cells mostly induced in the periphery (pTregs)[18,19] (Fig. 1L), both critical to maintain intestinal homeostasis[18–20,22,25]. In chow diet-fed animals, the proportion of RORγt⁺ pTregs was similar in the mesenteric lymph nodes (mesLNs) and small intestine, but markedly higher in the colon (Fig. 1M). Th17 effector T cells were more abundant in the small intestine and colonic mucosa as compared to mesLNs, where they are generated (Fig. 1N). In HFD-fed animals, the frequency of total Tregs was unaltered in the small intestine but slightly diminished in the colon (Fig. 1O), while RORγt⁺ pTregs waned at both sites (Fig. 1P). Similarly, RORγt⁺ Th17 cells were also decreased in the small intestinal and colonic lamina propria (Fig. 1Q). Absolute cell numbers were also reduced for both RORγt⁺ pTregs and Th17 cells in the small intestine and colon (Fig. S1A). Of note, the proportion of CD4 T cells, Tregs, and RORγt⁺ pTregs among CD45⁺ leukocytes decreased in both the small intestine and colon (Fig. S1B, C), while Th17 cells proportion was diminished in the small intestine only (Fig. S1C). Regarding Helios⁺ thymus-derived Tregs, their proportion was elevated in the colon while RORγt⁻ Helios⁻ double negative (DN) Tregs proportion increased in the small intestine and colon (Fig. S1D). By numbers, RORγt⁻ Tregs were reduced in the small intestine but raised in the colon (Fig. S1E), suggesting that the reduced proportion of RORγt⁺ Tregs in the colon is due to the concomitant decrease in RORγt⁺ pTregs and an increase in RORγt⁻ Tregs (Fig. S1A–E). Overall, we observed that RORγt⁺ CD4 T cell subsets were reduced in the small intestine and colon after 4 weeks of HFD. Similar observations were made when mice were fed the HFD for a longer period (14 weeks) (Fig. S1F–H).

We then focused on the thymus to assess whether those alterations could come from defects in thymopoiesis. After 4 weeks of HFD, thymus weight (Fig. S1I) and cellularity (Fig. S1J) did not change. Total Tregs and thymic T cell subsets (CD4⁺ CD8α⁺, CD4⁺ CD8α⁻, CD8α⁺ CD4⁻) were not altered either (Fig. S1K, L). We also did not observe any alteration in splenic T cell subsets (Fig. S1M–Q). Hence, the intestinal phenotype appears independent of impairments in T cell development in the thymus.

Changes in colonic and intestinal RORγt⁺ pTregs and Th17 cells could be due to obesity itself, changes in nutrients intake or both. We thus studied genetically obese Ob/Ob mice maintained on a chow diet or fed with the HFD. Within the small intestine, total Tregs were slightly increased in chow-fed Ob/Ob mice, while RORγt⁺ pTregs and Th17 cells were similar to lean, chow-fed controls (Fig. S1R). However, and comparable to WT mice, HFD decreased the frequency of both RORγt⁺ pTregs and Th17 cells in the small intestine (Fig. S1R). In the colon, independently of the genetic background or the diet, total Tregs remained unchanged (Fig. S1S). While colonic RORγt⁺ pTregs and Th17 were significantly reduced in chow-fed obese Ob/Ob, HFD further decreased RORγt⁺ pTregs (Fig. S1S). The dysbiosis that has been reported in Ob/Ob mice[26] might explain why colonic Th17 cells are already impaired under CD, although we did not measure it in our study. Overall, these observations suggest that both diet and body weight impact on RORγt⁺ CD4 T cells homeostasis, but that the diet has an effect per se.

Altogether, our observations reveal that 4 weeks of HFD feeding are sufficient to negatively impact on systemic glucose homeostasis and profoundly reduces RORγt⁺ pTregs and Th17 cells in the small intestine and colon.

### Fermentable dietary fructooligosaccharides supplementation improves glucose tolerance and prevents RORγt⁺ pTregs and Th17 cells loss in HFD-fed animals

Next, we sought to identify the nutritional signals participating in the immune and metabolic alterations induced upon HFD feeding. In the HFD, increased fat content is achieved at the expense of cereal starches rich in dietary fibers[10]. Recent studies pointed out the lack of fermentable dietary fibers as a leading cause of the dysbiosis and

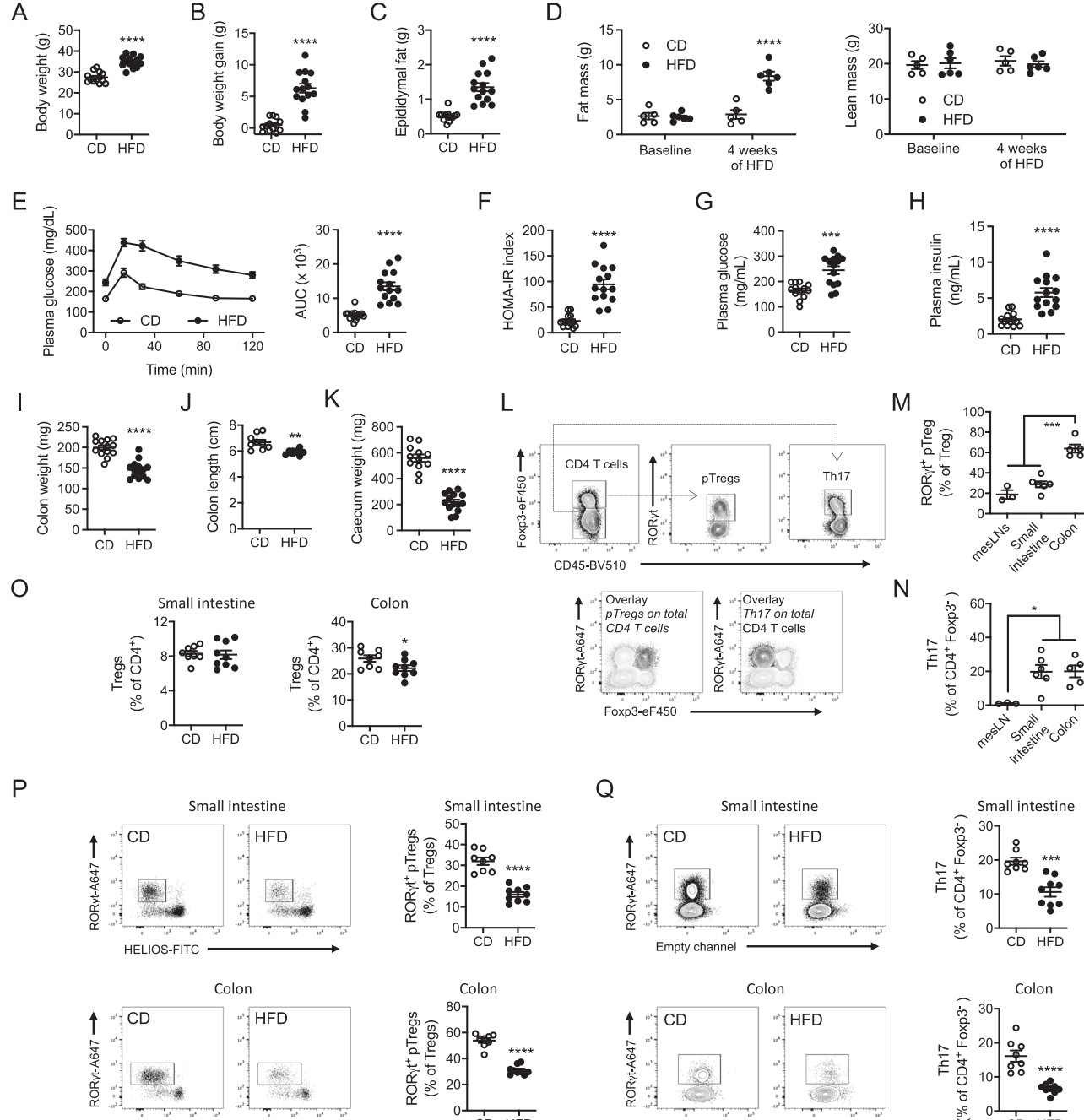

**Fig. 1 | HFD feeding decreases RORγt⁺ pTregs and Th17 cells in the small intestine and colon. A–D** Body weight ($n = 13–14$ mice per group) (**A**), body weight gain ($n = 13–14$ mice per group) (**B**), epididymal fat mass ($n = 13–14$ mice per group) (**C**) and body composition (fat and lean mass) ($n = 5–6$ mice per group, statistical significance assessed with 2-way ANOVA and Sidak's multiple comparison test, stars display adjusted $P$ value) (**D**) in wild-type mice fed a chow diet (CD) or a high-fat diet (HFD) for 4 weeks. **E, F** Oral glucose tolerance test and associated area under the curve (AUC) quantification (**E**), HOMA-IR index measurement (**F**), fasted plasma glucose (**G**), and insulin (**H**) levels in wild-type mice fed a chow diet (CD) or a high-fat diet (HFD) for 4 weeks ($n = 13–14$ mice per group). **I–K** Colon weight ($n = 13–14$ mice per group) (**I**), colon length ($n = 9–10$ mice per group) (**J**) and cecum weight ($n = 13–14$ mice per group) (**K**) in wild-type mice fed a chow diet (CD) or a high-fat diet (HFD) for 4 weeks. **L** Flow cytometry plots depicting RORγt⁺ CD4 T cells, including RORγt⁺ pTregs and Th17 cells, in the small intestine of chow diet-fed wild-type animals. **M, N** Flow cytometry analysis of RORγt⁺ pTregs (**M**) and Th17 cells (**N**) in the mesenteric lymph nodes (mesLNs) ($n = 3$ mice), the small intestine ($n = 6$ mice) and the colon ($n = 5$ mice) of chow diet-fed wild-type animals. Statistical significance tested with 1-way ANOVA and Newman-Keuls multiple comparison test. **O–Q** Flow cytometry analysis of total Tregs (**O**), RORγt⁺ pTregs (**P**) and Th17 cells (**Q**) in the small intestine and colon of wild-type mice fed a chow diet (CD) or a high-fat diet (HFD) for 4 weeks ($n = 8–9$ mice per group). All data in this figure are presented as mean values ± SEM. All panels correspond to two independent experimental groups. Statistical significance has been assessed with a two-sided $T$ test unless otherwise stated on the corresponding panel legend.

metabolic alterations associated with HFD feeding[10,11,27]. Since dietary fibers were shown to beneficially impact on immune homeostasis and the microbiota[28], we wondered whether their administration to HFD-fed animals would be sufficient to preserve RORγt⁺ pTregs and Th17

cells. In order to preserve HFD formulation, we administered fructooligosaccharides (FOS) as a fermentable fiber source in the drinking water. Under these conditions, FOS supplementation did not change body weight (Fig. 2A) nor significantly impacted body weight gain

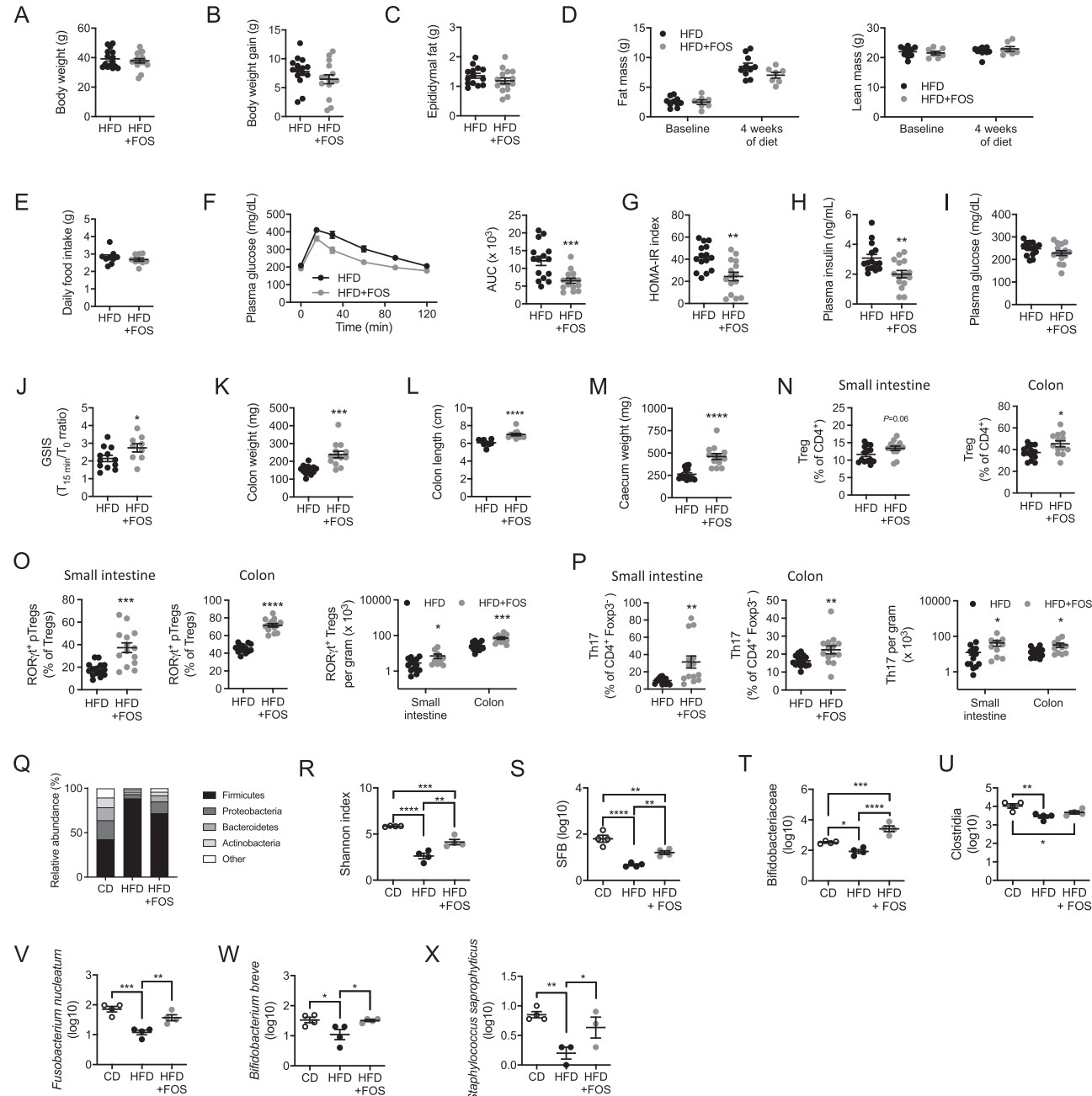

**Fig. 2 | Fructooligosaccharides supplementation improves glucose tolerance and prevents RORγt⁺ pTregs and Th17 cells loss in HFD-fed animals. A–E** Body weight (*n* = 15 mice per group) (**A**), body weight gain (*n* = 15 mice per group) (**B**), epididymal fat mass (*n* = 14–15 mice per group) (**C**), and body composition (fat and lean mass) (*n* = 7–10 mice per group. Statistical significance assessed with 2-way ANOVA and Sidak's multiple comparison test, stars display adjusted *P* value) (**D**) and daily food intake (*n* = 9 mice per group) (**E**) in wild-type mice fed a high-fat diet (HFD) or a high-fat diet supplemented with fructooligosaccharides (FOS) (HFD + FOS) administered in the drinking water for 4 weeks. **F–J** Oral glucose tolerance test and associated area under the curve (AUC) quantification (*n* = 15 mice per group) (**F**), HOMA-IR index measurement (*n* = 15 mice per group) (**G**), plasma insulin levels (*n* = 15 mice per group) (**H**), fasted plasma glucose levels (*n* = 15 mice per group) (**I**) and glucose-stimulated insulin secretion (GSIS) (*n* = 9–12 mice per group, 2 independent experimental groups) (**J**) in wild-type mice fed a high-fat diet (HFD) or a high-fat diet supplemented with FOS (HFD + FOS) for 4 weeks. **K–M** Colon weight (*n* = 13–14 mice per group) (**K**), colon length (*n* = 9–12 mice per group) (**L**), and cecum weight (*n* = 13–14 mice per group, two independent experimental groups)

(**M**) in wild-type mice fed a high-fat diet (HFD) or a high-fat diet supplemented with FOS (HFD + FOS) for 4 weeks. **N–P** Flow cytometry analysis of total Tregs (**N**), RORγt⁺ pTregs (**O**), and Th17 cells (**P**) in the small intestine and colon of wild-type mice fed a high-fat diet (HFD) or a high-fat diet supplemented with FOS (HFD + FOS) for 4 weeks (*n* = 13–14 mice per group). **Q–X** Microbiome sequencing and analysis of phyla relative abundance (**Q**), Shannon index of microbial diversity (**R**), relative abundance of specific bacterial class or species including Segmented Filamentous Bacteria (SFB) (**S**), Bifidobacteria (**T**), Clostridia (**U**), *Fusobacterium nucleatum* (**V**), *Bifidobacterium breve* (**W**), and *Staphylococcus saprophyticus* (**X**) in wild-type mice fed a chow diet (CD), a high-fat diet (HFD) or a high-fat diet supplemented with FOS (HFD + FOS) for 4 weeks (*n* = 4 mice per group, each from an independent experimental group. Statistical testing with one-way ANOVA and Newman-Keuls multiple comparison test). All data in this figure are presented as mean values ± SEM. All panels correspond to three independent experimental groups unless otherwise specified in the corresponding panel legend. Statistical significance has been assessed with a two-sided *T* test unless otherwise stated in the corresponding panel legend.

(Fig. 2B), epididymal fat mass (Fig. 2C), body composition (Fig. 2D), and food intake (Fig. 2E). Importantly, glucose tolerance was significantly improved by FOS supplementation (Fig. 2F). This was associated with a lowered HOMA-IR index (Fig. 2G) in FOS-treated animals that reflected decreased circulating insulin levels (Fig. 2H) rather than reduced plasma glucose concentrations (Fig. 2I). In addition, we observed that FOS increased glucose-stimulated insulin secretion (GSIS), suggesting improved pancreatic β-cell function (Fig. 2J). In summary, FOS supplementation improves systemic glucose homeostasis in HFD-fed animals.

The beneficial effect of FOS supplementation also translated into increased colon weight (Fig. 2K) and length (Fig. 2L) as well as cecum weight (Fig. 2M), reaching values similar to regular chow-fed animals (Fig. 1I–K). Beyond these morphologic parameters, FOS supplementation in HFD-fed mice did not significantly impact total Tregs in the small intestine, while it increased them in the colon (Fig. 2N). When expressed as the proportion of CD45[+] leukocytes, CD4 T cells and total Tregs, increased in the colon but not the small intestine (Fig. S2A, B). More specifically, FOS supplementation prevented the HFD-induced drop in RORγt[+] pTregs (Figs. 2O and S2A, B) and Th17 cells proportion and numbers (Figs. 2P and S2A, B) in the small intestine and colon. RORγt[-] Tregs numbers were also increased (Fig. S2C). Importantly, RORγt[+] pTregs and Th17 homeostasis were not affected by 4 weeks of HFD feeding nor FOS supplementation in other metabolic organs such as the liver (Fig. S3A) and the perigonadal adipose tissue (Fig. S3B). Thus, 4 weeks of HFD were not sufficient to increase Th17 cells in the adipose tissue and liver, which is known to have a detrimental impact on metabolic pathology after longer period of HFD feeding (superior to 12 weeks)[29–33]. Hence, altered RORγt[+] T cells homeostasis seems restrained to the intestine after 4 weeks of HFD.

Dietary fibers and their deprivation are capital in shaping intestinal bacterial ecology, which intricately dialogs with the immune system[8]. As RORγt[+] pTregs and Th17 cells development depends on the microbiota[18,34], we asked whether changes in microbial ecology could explain why RORγt[+] T cell subsets are altered in HFD-fed animals. As expected, the relative abundance of phyla switched towards an increase of the Firmicutes after HFD feeding (Figs. 2Q and S2D). As a consequence, microbiota diversity was drastically diminished by the HFD as measured by the Shannon index (Fig. 2R). FOS supplementation was able to partially correct, but not fully restore, Firmicutes over-representation and the loss of microbiota diversity (Figs. 2Q, R and S2D). Thus, HFD feeding-induced dysbiosis was partially corrected by FOS supplementation. RORγt[+] pTregs and Th17 cells are induced by specific genres and taxa of the microbiota[18,20,35,36]. SFB, the best-known Th17 inducer bacteria[20], was reduced upon HFD and increased following FOS supplementation (Fig. 2S). At the genre level, we assessed Bifidobacteria prevalence, which is a hallmark of fiber administration[27,37] and has been associated with Th17 cells generation[35]. We found that Bifidobacteria were decreased by the HFD and increased in FOS-supplemented animals (Fig. 2T). On the other hand, the most classically known inducers of RORγt[+] pTregs are Clostridia strains[36]. We observed that Clostridia were reduced in HFD-fed mice but FOS did not restore them (Fig. 2U). We also looked at specific strains previously shown to induce the generation of RORγt[+] pTregs in germ-free mice[18]. Among them, three strains, *Fusobacterium nucleatum, Bifidobacterium breve* and *Staphylococcus Saprophyticus* were reduced by the HFD and significantly increased by FOS (Fig. 2V–X). Overall, we show that FOS supplementation maintains several bacterial species and genres important for the generation of RORγt[+] pTregs and/or Th17 cells.

Altogether, we demonstrate that FOS supplementation in HFD-fed animals has a beneficial impact on glucose metabolism, prevents the loss of RORγt[+] pTregs and Th17 cells and limits the decrease in key microbial species supporting the development of RORγt[+] CD4 T cell subsets.

## Th17 generation and gut-homing imprinting in the mesenteric lymph nodes are impaired by HFD feeding and corrected upon FOS supplementation

The microbiota changes presented above suggests that the decrease in RORγt[+] CD4 T cells in the intestinal tract of HFD-fed mice could result from impaired priming and development in response to environmental cues. It could also be potentiated by reduced migration to the intestinal tract due to altered gut-homing imprinting. We thus focused on the mesenteric lymph nodes (mesLNs) where RORγt[+] pTregs and Th17 cells are generated and educated. While total Tregs (Fig. 3A) and RORγt[+] pTregs (Fig. 3B) were unaltered in the mesLNs of HFD-fed animals, Th17 cells were reduced (Fig. 3C). This revealed that Th17 cells priming was decreased upon HFD and this was associated with the drop in SFB abundance observed upon HFD feeding (Fig. 2S). We next evaluated the gut-homing imprinting of the two RORγt[+] T cell subsets. The frequency of RORγt[+] pTregs and Th17 cells expressing the gut-homing receptor CCR9 was reduced in HFD-fed mice (Fig. 3D, E). In addition, RORγt[+] pTregs expressing the gut-homing integrin α4β7 also decreased in the mesLNs upon HFD (Fig. 3D, E). Together, this indicates that the decline in intestinal Th17 cells observed after HFD feeding is owed to both decreased generation and gut-homing imprinting in the mesLNs, while RORγt[+] pTregs diminution would solely rely on defective gut-homing imprinting in the mesLN.

We then sought to decipher whether the FOS supplementation would maintain RORγt[+] T cells priming and gut-homing imprinting in the mesLNs of HFD-fed animals. FOS supplementation during HFD feeding increased total Tregs (Fig. 3F) as well as RORγt[+] pTregs and Th17 cells (Fig. 3G) in the mesLNs. In addition, CCR9[+] and ITGβ7[+] RORγt[+] pTregs (Fig. 3H) as well as CCR9[+] and ITGβ7[+] Th17 cells (Fig. 3I) were increased after FOS supplementation. To know whether this effect was specific to the HFD context, we supplemented chow diet (CD)-fed animals with FOS for 4 weeks. Total Tregs as well as RORγt[+] pTregs and their gut-homing imprinting were unaltered by the FOS supplementation upon CD (Fig. S2E). Th17 cells display a tendency to increase and CCR9[+] Th17 cells were significantly augmented (Fig. S2F). These results suggest that FOS are less efficient when administered on top of a fiber-rich CD. Thus, FOS exerts its beneficial effects when it complements the fiber-poor HFD.

FOS are fermented by the intestinal microbiota into short-chain fatty acids (SCFAs). We then asked whether FOS-mediated prevention of RORγt[+] T cells alterations in HFD-fed animals was due to their fermentation into SCFAs by the microbiota. To address this point, we administered acetate, butyrate and propionate in the drinking water during the 4 weeks of HFD feeding. We observed that SCFA supplementation increased Th17 cells and their gut- imprinting (Fig. 3J, K) while RORγt[+] pTregs were not modified (Fig. 3L, M). Thus, SCFAs were capable to limit the alterations in Th17 cells generation and education observed upon HFD feeding, but did not improve RORγt[+] pTregs gut-homing imprinting.

Overall, HFD feeding decreases Th17 cells generation and gut-homing imprinting in the mesLNs, while RORγt[+] pTregs only show an impairment of their gut-homing imprinting. In this context, FOS supplementation prevents the alterations observed in both RORγt[+] CD4 T cell subsets.

## IRF4-dependent dendritic cells (cDC2) participate in the homeostasis of RORγt[+] pTregs and Th17 cells in chow-fed animals

CD103[+] cDCs control intestinal T cells polarization in the mesLNs[38]. This DC subset is heterogeneous and can be further subdivided into CD103[+] CD11b[-] cDC1 and CD103[+] CD11b[+] cDC2[39]. CD103[-] CD11b[+] cDC2 are also found in the mesLN[40] but they remain less well characterized. While cDC1 development depends on the transcription factors BATF3 and IRF8[41,42], cDC2 rely on the transcription factor IRF4[43,44] and were shown to be instrumental in the induction of intestinal Th17 cells[43,44]. Which DC subset is responsible for the induction of intestinal RORγt[+]

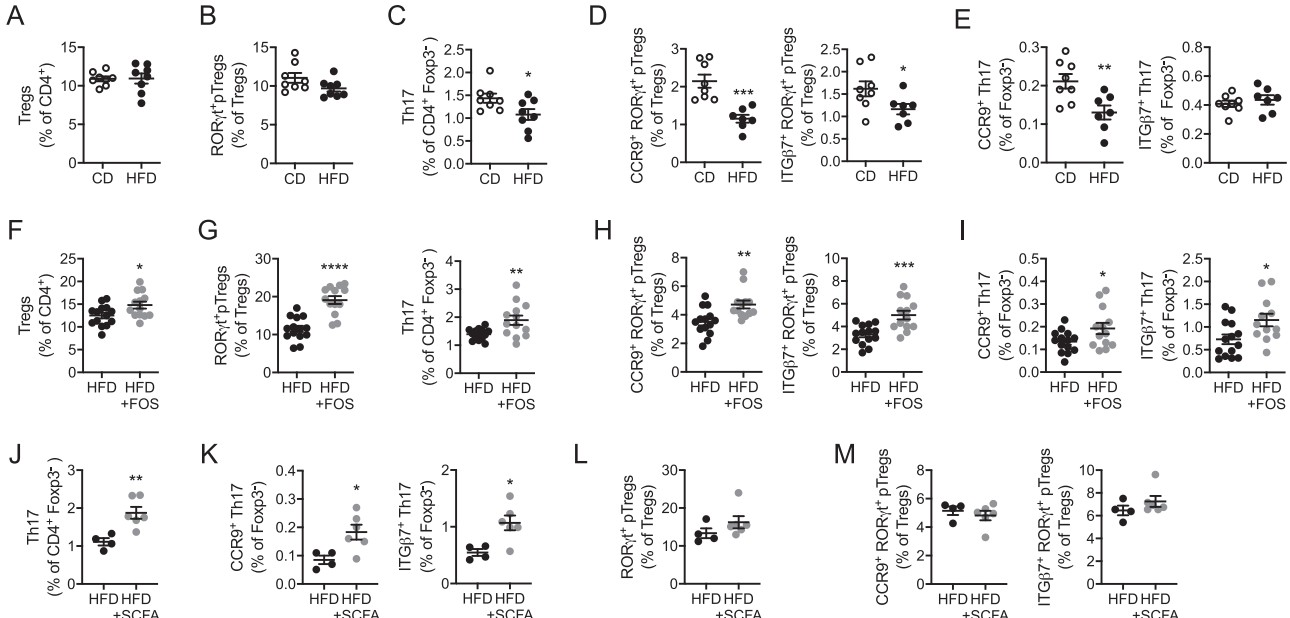

**Fig. 3 | Th17 cells generation and gut-homing imprinting of RORγt+ pTregs and Th17 cells are impaired by HFD feeding and corrected upon FOS supplementation.** A−C Flow cytometry analysis of total Tregs (**A**), RORγt+ pTregs (**B**), and Th17 cells (**C**) in the mesenteric lymph nodes of wild-type mice fed a chow diet (CD) or a high-fat diet (HFD) for 4 weeks (*n* = 8 mice per group, 2 independent experimental groups). **D, E** Flow cytometry analysis of RORγt+ pTregs (**D**) and Th17 cells (**E**) expressing CCR9 or ITGβ7 in the mesenteric lymph nodes of wild-type mice fed a chow diet (CD) or a high-fat diet (HFD) for 4 weeks (*n* = 7−8 mice per group, 2 independent experimental groups). **F, G** Flow cytometry analysis of total Tregs (**F**), RORγt+ pTregs (**G**), and Th17 cells (**G**) in the mesenteric lymph nodes of wild-type mice fed a high-fat diet (HFD) or a high-fat diet supplemented with FOS (HFD + FOS) for 4 weeks (*n* = 13−14 mice per group, 3 independent experimental groups). **H, I** Flow cytometry analysis of CCR9 or ITGβ7-expressing RORγt+ pTregs (**H**) and

Th17 cells (**I**) in the mesenteric lymph nodes of wild-type mice fed a high-fat diet (HFD) or a high-fat diet supplemented with FOS (HFD + FOS) for 4 weeks (*n* = 13−14 mice per group, 3 independent experimental groups). **J, K** Flow cytometry analysis of Th17 cells (**J**) and Th17 cells expressing CCR9 or ITGβ7 (**K**) in the mesenteric lymph nodes of wild-type mice fed a high-fat diet (HFD) or a high-fat diet supplemented in SCFA (HFD + SCFA) for 4 weeks (*n* = 4−6 mice per group, 1 experimental group). **L, M** Flow cytometry analysis of RORγt+ pTregs (**L**) and RORγt+ pTregs expressing CCR9 or ITGβ7 (**M**) in the mesenteric lymph nodes of wild-type mice fed a high-fat diet (HFD) or a high-fat diet supplemented in SCFA (HFD + SCFA) for 4 weeks (*n* = 4−6 mice per group, 1 experimental group). All data in this figure are presented as mean values ± SEM. Statistical significance has been assessed with a two-sided *T* test, stars display *P* value.

pTregs in adulthood remains, so far, less clear. Given that RORγt+ pTregs and Th17 cells share the expression of RORγt and are both altered upon HFD, we wondered if both subsets were primed by cDC2. In chow diet-fed *Itgax*-cre x *Irf4*^flox/flox^ (*Irf4*^ΔDC^) mice, CD103+ CD11b+ cDC2 were profoundly reduced in the intestinal lamina propria and mesLNs (Fig. S4A, B), as previously reported[43,44]. CD103− CD11b+ cDC2 were slightly reduced in the mesLNs (Fig. S4B) but not the lamina propria (Fig. S4A). cDC1 and macrophages were unaltered in the lamina propria (Fig. S4A) while cDC1 were slightly increased in the mesLN (Fig S4B). We found that both Th17 cells (Fig. 4A) and RORγt+ pTregs (Fig. 4B) were decreased in the small intestine, colon and mesLNs of *Irf4*^ΔDC^ mice as compared to controls. In addition, the gut-homing imprinting of RORγt+ CD4 T cell subsets was decreased in *Irf4*^ΔDC^ mice (Fig. 4C, D). In order to assess whether cDC1 would participate in RORγt+ pTregs priming at the steady state, we used chow-fed *Itgax*-cre x *Irf8*^flox/flox^ (*Irf8*^ΔDC^) mice, which lack CD103+ CD11b− cDC1 in the intestine (Fig. S5A) and mesLNs (Fig. S5B). In the small intestine, colon, and mesLNs, loss of cDC1 had no impact on RORγt+ pTregs (Fig. S5C) while Th17 cells were increased (Fig. S5D). Thus, cDC2, but not cDC1, participates in the maintenance of both Th17 cells and RORγt+ pTregs at steady state.

We then asked whether cDC subsets were affected upon HFD feeding and FOS supplementation. In HFD-fed animals, we observed a decrease in the number of CD103+ CD11b+ cDC2 in the mesLNs, while CD103− CD11b+ cDC2 and CD103+ CD11b− cDC1 remained unchanged (Fig. S2G). However, FOS supplementation couldn't increase cDC2 numbers in HFD-fed animals (Fig. S2H). This suggests that FOS supplementation prevented HFD-induced loss of RORγt+ pTregs and Th17 cells independently from the restoration of cDC2 numbers.

Gut-homing imprinting of CD4 T cells depends on retinoic acid (RA) produced by CD103+ cDCs in the mesLN through the action of aldehyde dehydrogenases (ALDHs) on vitamin A-derived retinol[45]. HFD feeding increased ALDH activity in both cDC1 and cDC2 (Fig. S2I), while FOS supplementation decreased it (Fig. S2J). Together, these observations indicate that the HFD did not induce intrinsic cDC2 dysfunction. It rather suggests that environmental cues, such as a limited retinol bioavailability that was previously reported in HFD-fed animals[46], could impact cDCs function including their gut-homing imprinting capacities.

Collectively, we show that cDC2 are responsible for the maintenance of both RORγt+ CD4 T cell subsets at steady state, and that cDC2 numbers are not impacted upon FOS supplementation.

## cDC2 control FOS-mediated prevention of Th17 cells loss and glucose tolerance improvement in HFD-fed animals

We showed above that cDC2 numbers are not affected by the FOS supplementation (Fig. S2H), but that cDC2 control both Th17 and RORγt+ pTregs at the steady state. We also showed that FOS supplementation prevented the HFD-induced decrease in microbial species known to stimulate RORγt+ CD4 T cells generation. Thus, we reasoned that cDC2 could control the benefits of FOS by sensing and mediating microbial-derived signals.

To test for the importance of cDC2 in mediating the benefits of FOS supplementation, we fed separated cohorts of *Irf4*^ΔDC^ mice and *Irf4*^flox/flox^ littermate controls with the HFD for 4 weeks and supplemented part of them with FOS. First, we observed that *Irf4*^ΔDC^ and *Irf4*^flox/flox^ control mice responded to FOS supplementation by increasing their colon weight and length as well as cecum weight

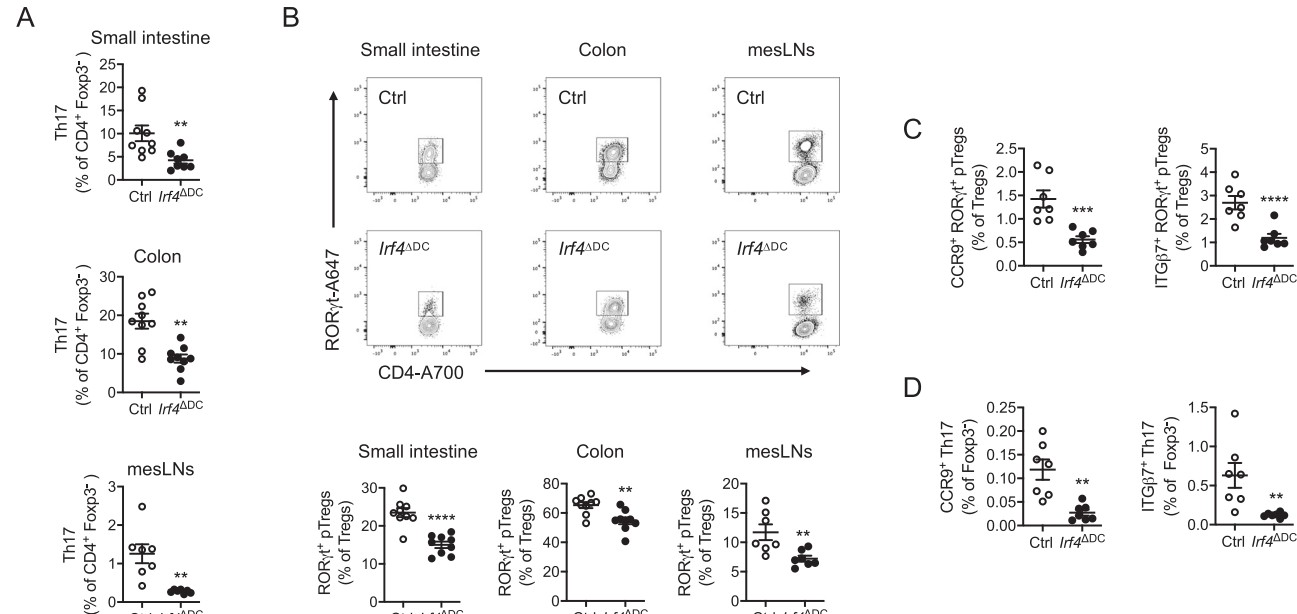

**Fig. 4 | IRF4-dependent dendritic cells (cDC2) participate to the homeostasis of RORγt⁺ pTregs and Th17 cells. A** Flow cytometry analysis of Th17 cells in the small intestine ($n = 9$ mice per group), colon ($n = 9$ mice per group) and mesenteric lymph nodes (mesLNs) ($n = 7$ mice per group) of mice lacking *Irf4* in dendritic cells (*Irf4*^ΔDC) and *Irf4*^flox/flox controls (ctrl). **B** Flow cytometry plots and analysis of RORγt⁺ pTregs in the small intestine ($n = 9$ mice per group), colon ($n = 9$ mice per group) and mesenteric lymph nodes (mesLNs) ($n = 7$ mice per group) of mice lacking *Irf4* in dendritic cells (*Irf4*^ΔDC) and *Irf4*^flox/flox controls (Ctrl). **C, D** Flow cytometry analysis of CCR9 or ITGβ7-expressing RORγt⁺ pTregs (C) and Th17 cells (D) in the mesenteric lymph nodes of mice lacking *Irf4* in dendritic cells (*Irf4*^ΔDC) and *Irf4*^flox/flox controls (Ctrl) ($n = 7$ mice per group). All data in this figure are presented as mean values ± SEM. Statistical significance has been assessed with a two-sided *T* test and stars display *P* value. All panels correspond to two independent experimental groups.

(Fig. 5A–C) as reported above for wild-type animals (Fig. 2K–M). We next turned our attention to RORγt⁺ CD4 T cell subsets. Unexpectedly, RORγt⁺ pTregs were increased upon FOS supplementation in the small intestine, colon, and mesLNs of mice lacking cDC2 to a comparable extent as in *Irf4*^flox/flox control animals (Fig. 5D–F). In addition, the lack of cDC2 had no impact on FOS-induced gut-homing imprinting of RORγt⁺ pTregs in the mesLNs (Fig. 5G, H). This suggests that other antigen-presenting cells were capable to compensate for the lack of cDC2 in this context. Nevertheless, while FOS increased Th17 cells in the small intestine, colon and mesLNs of *Irf4*^flox/flox control animals, their effect was blunted in *Irf4*^ΔDC mice lacking cDC2 (Fig. 5I–K). Moreover, Th17 cells gut-homing imprinting was not improved in FOS-supplemented *Irf4*^ΔDC mice as compared to their littermate *Irf4*^flox/flox controls (Fig. 5L, M). Thus, while FOS-mediated prevention of RORγt⁺ pTregs loss most likely benefit from cDC2-independent mechanisms, FOS impact on Th17 cells was fully dependent on cDC2. We then wondered if the absence of cDC2 had any impact on the beneficial metabolic effects of FOS supplementation in HFD-fed animals. To this aim, separated cohorts of *Irf4*^ΔDC mice and *Irf4*^flox/flox littermate controls were fed the HFD for 4 weeks and part of the animals were supplemented with FOS. Separated cohorts were studied as we could not perform the metabolic exploration simultaneously with such a number of animals. Body weight (Fig. 6A), weight gain (Fig. 6B), epididymal fat mass (Fig. 6C), body composition (Fig. 6D), and food intake (Fig. 6E) were not significantly modified by FOS supplementation in *Irf4*^ΔDC and *Irf4*^flox/flox littermate controls. Importantly, while FOS improved glucose tolerance in *Irf4*^flox/flox control animals, this effect was abrogated in *Irf4*^ΔDC mice lacking cDC2 (Fig. 6F), reflecting the lower HOMA-IR index (Fig. 6G) and better glucose-stimulated insulin secretion (Fig. 6H) of *Irf4*^flox/flox animals but not *Irf4*^ΔDC mice.

We then asked whether FOS requires an intact IL-17 signaling to improve glucose tolerance in HFD-fed mice. To answer that question, we treated HFD-fed animals supplemented with FOS with neutralizing antibodies directed against IL-17A and IL-17F, which are both produced

by Th17 cells, or their appropriate isotype control. While IL-17 neutralization did not impact on body fat mass (Fig. 6I), it limited the beneficial impact of FOS supplementation on glucose clearance (Fig. 6J). Thus, the beneficial metabolic impact of FOS partially depends on IL-17 to ameliorate glucose homeostasis, arguing that a cDC2-Th17 axis participates to the FOS-mediated improvement of metabolic fitness in HFD-fed animals.

Finally, we investigated how cDC2 could prevent HFD-induced impairments in glucose homeostasis in response to FOS. We first probed the endocrine function of the intestine. Indeed, FOS were previously reported to regulate the GLP1-GLP1R axis and the mRNA expression of proglucagon (*Gcg*, encoding the GLP-1 precursor) in the intestine to regulate glucose homeostasis in mice[47]. We observed a marked trend towards decreased *Gcg* mRNA expression in the colon of HFD-fed mice, its major expression site (Fig. S6A). In addition, the HFD reduced colonic *Pyy* mRNA expression (Fig. S6A), which encodes an intestinal peptide capable to improve glucose tolerance[48]. While FOS supplementation did not modulate *Gcg* expression, it increased *Pyy* levels in *Irf4*^flox/flox control mice but not in *Irf4*^ΔDC animals (Fig. S6B). We then assessed the effects of cDC2 depletion on the intestinal barrier upon FOS supplementation. Indeed, the barrier function of the intestine has been suggested to be instrumental in maintaining metabolic health[49,50]. After 4 weeks of HFD, we observed a deregulation of genes encoding key antimicrobial peptides (*Reg3b* and *Reg3g*) (Fig. S6C). In addition, the mRNA levels of the genes coding for the main proteins forming the mucus layer (*Muc2* and *Muc3*) (Fig. S6D) were decreased. FOS supplementation increased *Reg3b*, *Reg3g*, *Muc2*, and *Muc3* in *Irf4*^flox/flox control mice but not in *Irf4*^ΔDC animals (Fig. S6E). Those results suggest that cDC2 also channels the metabolic benefits of FOS by contributing to the maintenance of the intestinal barrier function. As RegIII family lectins and mucins maintain a buffering layer between the host and the microbiota, FOS may act through a cDC2-dependent axis to restore beneficial physicochemical properties of the lumen allowing symbiosis between the microbiota and the epithelium. Along these

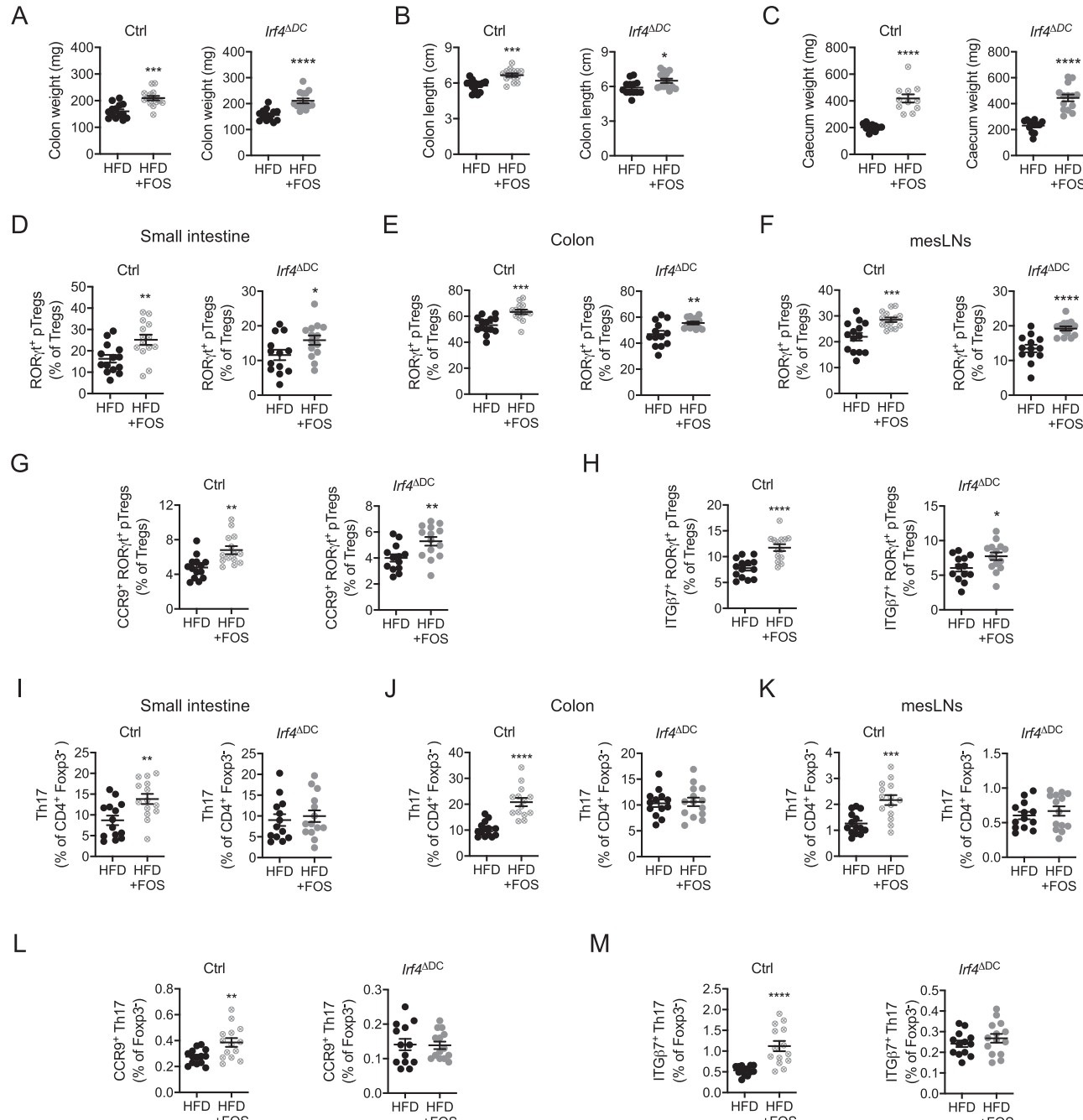

**Fig. 5 | cDC2 control FOS-mediated prevention of Th17 cells loss in HFD-fed animals. A–C** Colon weight (**A**), colon length (**B**), and cecum weight (**C**) in mice lacking *Irf4* in dendritic cells (*Irf4*ΔDC) (*n* = 13–14 mice per group) and *Irf4*flox/flox controls (ctrl) (*n* = 11–14 mice per group) fed a high-fat diet (HFD) or a high-fat diet supplemented with FOS (HFD + FOS) for 4 weeks. Panel **C** includes two independent experimental groups. **D–F** Flow cytometry analysis of RORγt⁺ pTregs in the small intestine (**D**), colon (**E**) and mesenteric lymph nodes (mesLNs) (**F**) of mice lacking *Irf4* in dendritic cells (*Irf4*ΔDC) (*n* = 13–14 per group) and *Irf4*flox/flox controls (ctrl) (*n* = 11–14 mice per group) fed a high-fat diet (HFD) or a high-fat diet supplemented with FOS (HFD + FOS) for 4 weeks. **G**, **H** Flow cytometry analysis of CCR9 (**G**) or ITGβ7 (**H**)-expressing RORγt⁺ pTregs in the mesenteric lymph nodes (mesLNs) of mice lacking *Irf4* in dendritic cells (*Irf4*ΔDC) (*n* = 13–14 mice per group) and *Irf4*flox/flox controls (ctrl) (*n* = 14 mice per group) fed a high-fat diet (HFD) or a

high-fat diet supplemented with FOS (HFD + FOS) for 4 weeks. **I–K** Flow cytometry analysis of Th17 cells in the small intestine (**I**), colon (**J**), and mesenteric lymph nodes (mesLNs) (**K**) of mice lacking *Irf4* in dendritic cells (*Irf4*ΔDC) (*n* = 13–14 mice per group) and *Irf4*flox/flox controls (ctrl) (*n* = 14 mice per group) fed a high-fat diet (HFD) or a high-fat diet supplemented with FOS (HFD + FOS) for 4 weeks. **L, M** Flow cytometry analysis of CCR9 (**L**) or ITGβ7 (**M**)-expressing Th17 cells in the mesenteric lymph nodes (mesLNs) of mice lacking *Irf4* in dendritic cells (*Irf4*ΔDC) (*n* = 13–14 mice per group) and *Irf4*flox/flox controls (ctrl) (*n* = 14 mice per group) fed a high-fat diet (HFD) or a high-fat diet supplemented with FOS (HFD + FOS) for 4 weeks. All data in this figure are presented as mean values ± SEM. Statistical significance has been assessed with a two-sided *T* test and stars display *P* value. All panels correspond to three independent experimental groups unless otherwise stated in the corresponding panel legend.

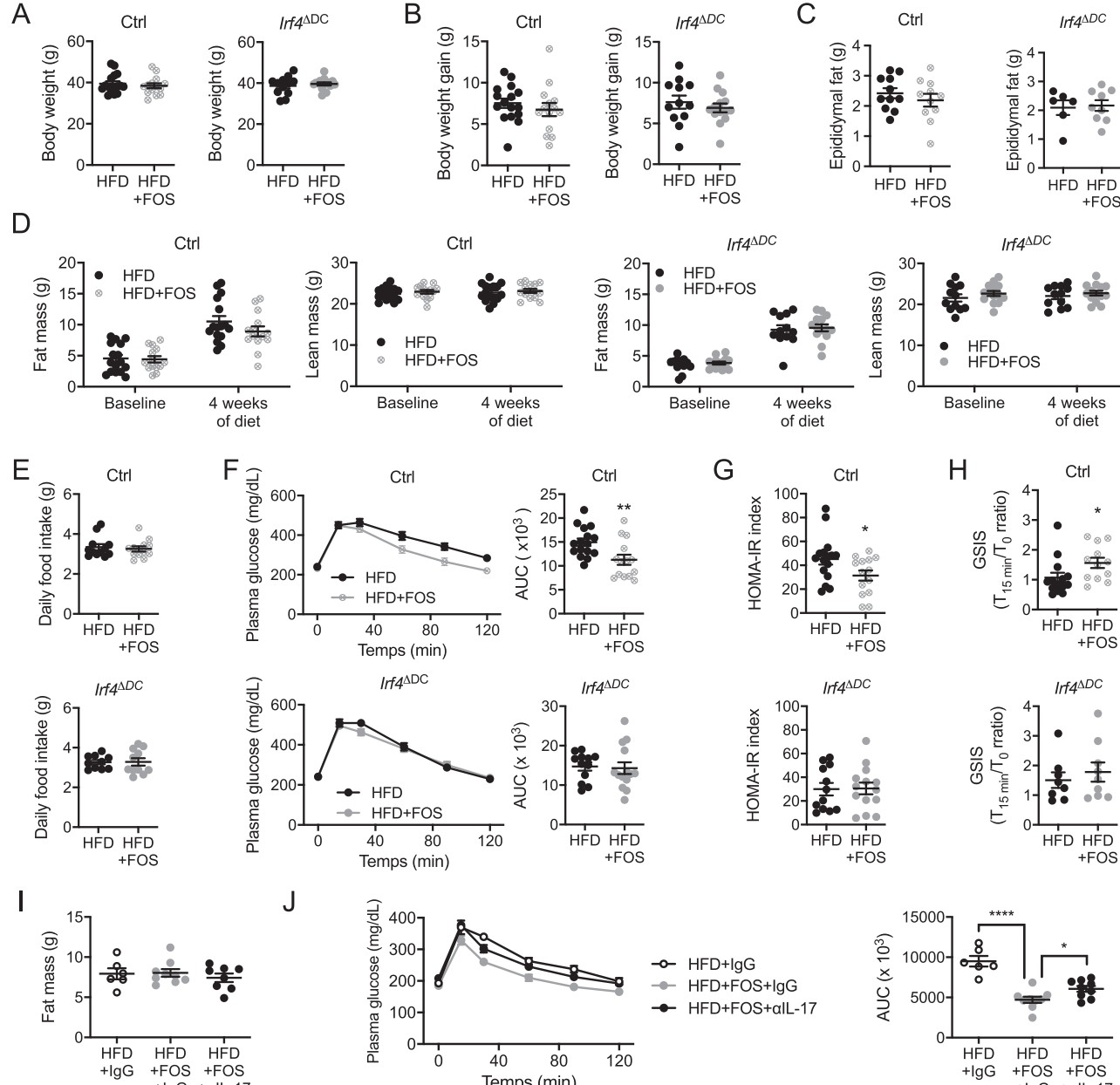

**Fig. 6 | cDC2 control FOS-mediated improvement of glucose tolerance in HFD-fed animals. A–E** Body weight (**A**), body weight gain (**B**), epididymal fat mass (**C**), body composition (fat and lean mass) (**D**), and daily food intake (**E**) in mice lacking *Irf4* in dendritic cells (*Irf4*$^{\Delta DC}$) ($n = 6$–14 mice per group) and *Irf4*$^{flox/flox}$ controls (ctrl) ($n = 11$–16 mice per group) fed a high-fat diet (HFD) or a high-fat diet supplemented with FOS (HFD + FOS) for 4 weeks. Panels **A**, **B** and **D** include three independent experimental groups, panel **C** includes two independent experimental group. Statistical significance in panel **D** was assessed with one-way ANOVA and Sidak's multiple comparison test and stars display adjusted *P* value. **F–H** Oral glucose tolerance test and associated area under the curve (AUC) quantification (**F**), HOMA-IR index measurement (**G**), and glucose-stimulated insulin secretion (GSIS) (**H**) in mice lacking *Irf4* in dendritic cells (*Irf4*$^{\Delta DC}$) ($n = 8$–14 mice per group) and *Irf4*$^{flox/flox}$ controls (ctrl) ($n = 12$–16 mice per group, panels **F** and **G** include three independent experimental groups, panel **H** includes two independent experimental groups) fed a high-fat diet (HFD) or a high-fat diet supplemented with FOS (HFD + FOS) for 4 weeks. **I, J** Body fat mass (**I**) and oral glucose tolerance test with the associated area under the curve (AUC) quantification (**J**) measured after 4 weeks of HFD in mice treated with an isotype control (HFD + IgG) as well as mice supplemented with FOS and treated with an isotype control (HFD + FOS+IgG) or antibodies neutralizing IL-17A and IL-17F (HFD + FOS + αIL-17) ($n = 6$–9 mice per group, two independent experimental groups, statistical significance tested with one-way ANOVA and Newman-Keuls multiple comparison test). All data in this figure are presented as mean values ± SEM. Statistical significance has been assessed with a two-sided *T* test unless stated otherwise on the corresponding panels.

lines, previous studies showed that Th17 cells-derived cytokines regulate antimicrobial peptides production[51–53]. In sum, we show that cDC2 are needed to mediate some positive impacts of FOS on intestinal functions known to be instrumental in glucose homeostasis. Several mechanisms likely add up in order to prevent the impairment of glucose tolerance.

Together, our work provides insight regarding the cellular mechanisms by which the fermentable dietary fiber fructooligo-saccharides beneficially impact glucose metabolism. Here, we uncovered a previously unappreciated role for cDC2 in mediating these fiber's effects on Th17 cells homeostasis and systemic glucose tolerance in the context of HFD feeding.

## Discussion

HFD-induced obesity remains the standard model to study obesity pathogenesis in preclinical models. HFD mimics the human "western" diet, reflecting increased fat content at the expense of dietary fibers. Such nutritional challenge leads to metabolic dysfunctions, including altered glucose homeostasis, and was shown to impact intestinal immune homeostasis. We investigated the effect of a short-term HFD regimen (4 weeks) and whether specific changes in intestinal immune subsets participate in the dysregulation of glucose homeostasis. Here, we report that HFD induces dysbiosis, including the reduction in bacterial strains known to favor RORγt+ CD4 T cell generation, and decreases intestinal RORγt+ pTregs and Th17 cells. Importantly, supplementation of HFD-fed animals with the fermentable dietary fiber fructooligosaccharides (FOS) was sufficient to prevent the drop in RORγt-inducing bacterial species, maintain both RORγt+ CD4 T cell subsets and amend glucose homeostasis in HFD-fed animals. The beneficial effect of FOS required cDC2 to preserve Th17 cells and improve glucose tolerance. Overall, our findings unveil a previously unappreciated role of type 2 conventional dendritic cells in mediating the beneficial impact of FOS intake on mucosal immunity and glucose homeostasis in the context of HFD feeding.

RORγt+ pTregs and Th17 cells were decreased in the small intestine and colon of HFD-fed animals, as it was previously shown for the whole RORγt+ CD4 T cell pool[17]. This previous observation was attributed to dysbiosis[17] as the gut microbiota plays a central role in the induction of RORγt expression in Th17 cells[34,54] and peripheral Tregs[18,19]. Recently, the HFD-mediated changes in gut microbiota ecology were shown to mostly stem from the low dietary fiber content[11]. We reveal here that FOS intake limited the HFD-mediated reduction in the Th17-inducing bacteria SFB[20] and the RORγt+ pTregs-inducing species B. breve and F. nucleatum[18]. FOS supplementation also prevented the decrease of intestinal RORγt+ pTregs and Th17 cells in HFD-fed animals, by maintaining their priming and/or gut-homing imprinting in the mesLNs. Overall, we report that nutritional imbalance, i.e. dietary fiber paucity, triggers dysbiosis and impairs the maintenance of RORγt+ pTregs and Th17 cells. A very recent study suggests that high-sugar content can also lead to reduced SFB abundance and intestinal Th17 cells upon HFD feeding[55]. This sugar-induced effect is dependent upon competition mechanisms with other members of the microbiota. Thus, different nutritional cues, such as the paucity of readily fermentable dietary fibers or an excess of sugar in the diet, favor an imbalance in microbiota species including decreased SFB abundance. The same study demonstrates that supplementing HFD with SFB improves both intestinal immunity and metabolic health upon 4 weeks of HFD, highlighting the importance of SFB decrease in our conditions.

The beneficial effect of FOS in restoring RORγt+ CD4 T cell subsets homeostasis likely relied on cDCs as both RORγt+ pTregs and Th17 cells need cDCs to develop[19,43,44]. First, consistent with the previously described role of Irf4-dependent cDC2 in Th17 cell generation[43,44], we found that FOS-mediated preservation of Th17 cells in HFD-fed animals required cDC2. Then, we observed that cDC2, but not cDC1, participated to RORγt+ pTregs priming at the steady state. Yet, FOS supplementation led to the maintenance of RORγt+ pTregs in HFD-fed cDC2-deficient animals. Even though cDC2 have a dominant impact on RORγt+ pTregs at the steady state, a significant proportion of RORγt+ pTregs remained in their absence. This suggests that cDC1 could somewhat compensate the absence of cDC2 at the steady state and explain why RORγt+ pTregs were retained upon FOS supplementation in HFD-fed cDC2-deficient animals. Plasmacytoid DCs (pDCs) could also play a role in RORγt+ pTregs homeostasis. Indeed, both cDC1, cDC2 and pDCs are lacking in Cd11c-cre x LsL-ROSA-DTA mice[56] in which RORγt+ pTregs do not develop[19], and pDCs were previously shown to favor RORγt+ pTreg generation in the intestinal tract[57]. A recent study suggests that different intestinal DC subsets, including CD103+ cDC1 and cDC2 as well as CD103- CD11b+ cDCs, have the ability to induce RORγt+ pTregs generation depending on the context[58]. Finally, other cell types such as ILC3[59] or RORγt+ antigen-presenting cells[60,61] may also be involved in this compensation. In summary, while FOS intake maintains RORγt+ pTregs homeostasis in the absence of cDC2, Th17 cells observe a strict dependency on cDC2 to benefit from FOS supplementation.

Since FOS did not fully maintain RORγt+ CD4 T cells in the absence of cDC2, we tested whether cDC2 deficiency altered FOS ability to improve glucose handling. We observed that FOS failed to improve glucose tolerance in HFD-fed mice lacking cDC2. We thus identified a key role for cDC2 in mediating the beneficial metabolic effects of FOS in HFD-fed animals. This effect did not depend on RORγt+ pTregs but partially relied on Th17 cells since they were not restored in FOS-treated cDC2-deficient animals and IL-17 neutralization limited the full impact of FOS on glucose tolerance. On their side, RORγt+ pTregs might control other aspects of intestinal homeostasis given their ability to prevent Th2-driven intestinal inflammation[19]. Importantly, while we focused our attention on particular subsets of CD4 T cells, other alterations in immune intestinal cells have been reported upon long-term HFD feeding[15,16], including an increase in Th1 IFNγ-producing CD4 T cells[62]. Since intestinal Th1 responses are controlled by cDC1[63], further work would be needed to assess the role of cDC1 in HFD-induced metabolic alterations. Together, we show cDC2 are critical to mediate the beneficial impact of FOS on Th17 cells homeostasis and glucose homeostasis in HFD-fed animals.

Fermentable dietary fibers were previously shown to protect against HFD-induced obesity. In these studies, adding fibers to the diet markedly limited weight gain and fat mass expansion, leading to an improvement in glucose tolerance[10,27]. Here, we delivered the fermentable dietary fiber FOS in the drinking water to keep the diet formulation similar between groups. Under our experimental conditions, FOS intake was capable to improve glucose handling independently from any major effect on adiposity. Improved glucose tolerance appeared independent from glucose absorption since blood glucose levels peaked similarly in FOS-treated animals and controls after oral glucose challenge. However, we observed that FOS supplementation accelerates glucose clearance together with improved glucose-stimulated insulin secretion. In this context, multiple mechanisms may concur to ameliorate glucose tolerance in FOS-treated animals.

In summary, the lack of fermentable dietary fiber is an important nutritional cue leading to decreased intestinal RORγt+ pTregs and Th17 cells in HFD-fed animals while cDC2 link FOS intake to Th17 homeostasis and the improvement of glucose tolerance. Overall, we provide insight regarding the cellular mechanisms by which fructooligosaccharides as a source of dietary fibers beneficially impact glucose metabolism. More specifically, we uncover a pivotal role of cDC2 in the control of the immune and metabolic effects of the fermentable dietary fiber fructooligosaccharides.

## Methods

### Mice and housing

Mice were housed in individually ventilated cages at a temperature of 22 °C and humidity of 50%, and maintained under specific pathogen-free conditions on a 12-h light and dark cycle with ad libitum access to water and diet (A04; Safe-Diets). Age-matched male mice were grouped by cages at weaning according to their genotype.

Wild-type C57BL/6J mice were from Charles River and bred in-house. Ob/+ (B6.Cg-Lep^ob/J) mice were from Charles River and bred in house to generate obese Ob/Ob mice and lean littermate controls (including Ob/+ and +/+ animals). Itgax-cre (B6.Cg-Tg(Itgax-cre)1-1Reiz/J), Irf4^flox/flox (B6.129S1-Irf4^tm1Rdf/J) and Irf8^flox/flox (B6(Cg)-Irf8^tm1.1Hm/J) were all obtained from the Jackson Laboratory. Itgax-cre mice were crossed to Irf4^flox/flox animals in our facility, while Itgax-cre x Irf8^flox/flox were directly imported from the Tussiwand lab (Basel Institute,

Switzerland). Mice between 10 to 14 weeks of age were used, littermate cre-negative mice were used as controls and germline deletion events were screened. The presence of a deleted *Irf4* allele was identified with the following primers: *Irf4* deleted forward – ccatggtggcgggatccaat and *Irf4* deleted reverse – cttcctcatctccgggcctttcg. This results in an approximate 120 bp band. The presence of a deleted *Irf8* allele was identified with the following primers: *Irf8* forward – caaaaaag-caggctggcgccg and *Irf8* deleted reverse – ccctttgaactgatggcgagctc. This results in an approximate 170 bp band.

All animal procedures were in accordance with the Guide for the Care and Use of Laboratory Animals published by the European Commission Directive 86/609/EEC and given authorization from the French Ministry of Research and local ethics committee (Charles Darwin, CEEA – 005).

### Diet and treatment
For high-fat diet (HFD)-induced metabolic dysfunctions studies, male mice (10–14-week-old) were fed a HFD in which 60% of kilocalories come from fat (D12492, Research Diets) and were compared to chow diet-fed animals (A04; Safe-Lab). The duration of feeding was indicated in the text and figure legends. For dietary fiber supplementation studies, mice were given fructooligosaccharides (obtained from Sigma-Aldrich or Beneo) diluted at 7.5% in filtered drinking water. The solution was renewed every 2 to 3 days for the entire diet duration. The short-chain fatty acids acetate, propionate, and butyrate were obtained from Sigma-Aldrich and diluted in drinking water at a concentration of 60 mM, 25 mM, and 40 mM, respectively. IL-17A (clone 17F3) and IL17F (clone MM17F8F5.1A9) neutralizing antibodies, as well as their appropriate isotype control (MOPC-21), were obtained from BioXcell. Antibodies (200 µg per injection) were injected intraperitoneally 3 times a week over the 4 week period of HFD feeding.

### Glucose metabolism assessment
For assessment of oral glucose tolerance, mice were fasted for 5 h prior to glucose intragastric gavage at a dose of 1.5 grams per kg of body weight. Glycaemia was measured with a glucometer (Accu-Check, Roche) at baseline and 15, 30, 60, 90, and 120 min after gavage. Blood was also collected at baseline and 15 min after gavage for insulin dosage. Insulin dosage was performed with the mouse ultrasensitive insulin ELISA kit from Alpco. The glucose-stimulation insulin secretion index was calculated as the ratio of blood insulin levels measured 15 min after glucose gavage to blood insulin levels at baseline. Finally, the HOMA-IR index was calculated with the following formula: fasting plasma insulin (mU/mL) × fasting plasma glucose (mm/L)/22.5.

### Fat and lean mass measurement
Fat and lean mass were measured by TD-NMR using a MinispecPlus LFII90 body composition analyzer (Bruker; PreclinICAN Plateform, Paris).

### Tissue processing and cell suspension preparation
For isolation of lamina propria leukocytes, freshly harvested intestines and colons were quickly washed in PBS, opened, and cut into smaller pieces. To remove epithelial cells, samples were placed into 40 mL of PBS (no calcium and magnesium) containing glucose (1 g/L), HEPES (10 mM), EDTA (5 mM), fetal bovine serum (5%), and dithiothreitol (0.5%), and incubated for 30 min at 37 °C under vigorous agitation. After cells were washed 5 times in 40 mL PBS, samples were chopped with scissors and placed in the digestion solution. The digestion solution consisted of HBSS (with calcium and magnesium) containing fetal bovine serum (3%), collagenase D (1.25 mg/mL, Sigma-Aldrich), DNase (10 U/mL, Sigma-Aldrich). Digestion was performed at 37 °C under agitation for 30 min. After completion, cell suspensions were passed through a 18G needle before filtration on a 70 µm filter, washed, and finally resuspended in PBS containing BSA (1%).

Liver and adipose tissue samples were directly minced into the digestion solution described above and then processed similarly to the intestines and colons.

Pooled mesenteric lymph nodes were cut open with a needle and digested as described above. Cell suspensions were eventually resuspended in PBS containing BSA (1%) before staining.

### Flow cytometry
Antibodies were purchased from BioLegend, ThermoFisher Scientific, and BD Biosciences. The following markers and clones were used: CD11c (N418), MHC-II (I-A/I-E, M5/114.15.2), CD103 (2E7), CD11b (M1/70), RORγt (Q31-378), Foxp3 (FJK-16s), CD4 (GK1.5), CCR9 (CW-1.2), ITGβ7 (DATK32), CD64 (X54-5/7.1) and CD45 (30-F11). Cell suspensions were stained with appropriate antibodies for 30 min on ice. Intracellular staining was performed using the Foxp3 staining kit from ThermoFisher Scientific. Aldehyde dehydrogenase (ALDH) activity was measured using the AldeRed™ ALDH Detection Assay (Merck Millipore) according to the manufacturer's instructions.

Data were acquired on a BD LSRFortessa™ flow cytometer (BD Biosciences) and analyzed with FlowJo software (Tree Star). To calculate absolute counts, a fixed number of non-fluorescent beads (10000, 10-µm polybead carboxylate microspheres from Polysciences) was added to each tube. The formula number of cells = (number of acquired cells × 10,000) / (number of acquired beads) was used. Cell counts were finally expressed as a number of cells per milligram of tissue.

### Microbiome sequencing and analysis
Fecal DNA was extracted using the NucleoMag DNA Microbiome kit (Macherey-Nagel) and sequenced using the MinION from Oxford Nanopore Technologies (ONT). The DNA library was prepared with the Ligation Sequencing Kit with multiplexing (ONT). The R studio software and the Nanopore.2.0 pipeline (https://git.ummisco.fr/ebelda/nanopore.v2.0) were used to analyze microbiome sequencing data[64].

### qPCR analysis of colon samples
Total RNA was extracted from frozen colon samples (20 mg) using the Nucleospin RNA Plus kit (Macherey-Nagel). cDNA was generated with the Transcriptor First strand cDNA Synthesis kit (Roche). Quantitative PCR was performed with SYBR Green I Master (Roche) on a Light-Cycler® 480 real-time PCR system with dedicated software (Roche). Gene expression was normalized to at least 2 housekeeping genes using the Roche LightCycler® 480 software.

The primers sequences are the following: *Gcg*, forward - tacacctgttcgcagctcag and reverse - ttgcaccagcattataagcaa; *Pyy*, forward - ttcgagcttctcccaccatt and reverse - cgagcaggattagcagcatt; *Reg3b*, forward - tggattgggctccatgac and reverse - tcatcacgtcattgttactcca; *Reg3g*, forward - accatcaccatcatgtcctg and reverse - ggcatctttcttggcaactt; *Muc2*, forward - acctccaggttcaacaccag and reverse - gttggccctgttgtgtgtct; *Muc3*, forward - agctgcagcgaagtggac and reverse - ccgctgtaccagtgagtatcc.

### Quantification and statistical analysis
Statistical significance of differences was performed using GraphPad Prism (GraphPad Software). Two-tailed Student's *t* test was used to assess the statistical significance of the difference between means of two groups. When stated, one-way or two-way ANOVA followed by multiple comparison tests were conducted. Graphs depicted the mean ± SEM. Statistical significance is represented as follows: $*P < 0.05$, $**P < 0.01$, $***P < 0.001$, and $****P < 0.0001$.

### Reporting summary
Further information on research design is available in the Nature Portfolio Reporting Summary linked to this article.

## Data availability

Data supporting the findings described in this manuscript are available in the article, in the Supplementary Information and from the corresponding author upon request. Source data are provided with the paper. Microbiota sequencing data are available on the Sequence Read Archive (SRA) (PRJNA1099178, https://www.ncbi.nlm.nih.gov/sra/PRJNA1099178). Source data are provided with this paper.

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

## Acknowledgements

This work was supported by grants to ELG from the Fondation de France (project number 00056835), the Agence Nationale pour la Recherche (ANR-17-CE14-0009, ANR-17-CE14-0023 and ANR-21-CE14-0023) and from the city of Paris (Emergence-s- program). Melissa Ouhachi received a one-year doctoral fellowship from the Nouvelle Société Française d'Athérosclérose (NSFA).

## Author contributions

A.G. provided intellectual input, designed and performed experiments, analyzed and interpreted data, and wrote the manuscript. G.M. provided intellectual input, designed and performed experiments, analyzed and interpreted data, and edited the manuscript. M.O., R.M., I.B., C.R., S.D., and L.V. performed experiments and analyzed data. L.Y.C., R.T., and K.C. provided intellectual input and edited the manuscript. T.H. provided intellectual input, designed experiments, and edited the manuscript. E.L.G. conceptualized and supervised the study, designed experiments, analyzed and interpreted data, and wrote the manuscript.

## Competing interests

The authors declare no competing interests.
