## [Peer Review File · Nature Communications]

Fructooligosaccharides benefits on glucose homeostasis upon high-fat diet feeding require type 2 conventional dendritic cells.REVIEWER COMMENTS

Reviewer #1 (Remarks to the Author):

In this manuscript, Gelineau et al investigated the impact of high fat diet and dietary fibre on small intestine and colonic Th17 and Treg and glucose homeostasis. They show that dietary fibre could rescue the effect of high fat diet on gut immunity and glucose homeostasis and that these effects involved type 2 dendritic cells. The manuscript is potentially interesting but too many mechanisms remain elusive. It would strengthen the manuscript if the following were addressed:

1. The absolute number of immune cells should be shown not just the proportion. It would also be good to gate the Th17 and Treg out of CD45+ cells not just CD4+ population.
2. What is the proportion of CD4+ T cells in HFD mice in the small intestine and colon as well as in the thymus to exclude a defect in thymopoiesis.
3. What is the proportion of RORgt- Treg in the colon and small intestine of HFD fed mice as well as in other sites such as the spleen and the proportion of thymic derived Helios+ Treg.
4. What are the blood and gut levels of short chain fatty acids in the HFD mice vs NC and HFD + FOS treated mice? This would confirm the higher gut microbial fermentation activity in FOS treated animals.
5. FOS has beneficial effects on metabolic and immune outputs potentially through the reshaping of the gut microbiota and/or via the microbial release of SCFA. To understand the mechanism the following should be addressed: 1) does the supplementation of drinking water with SCFA mimic the effects of FOS or 2) does the colonisation of GF mice with HFD microbiota vs HFD+FOS microbiota mimic the effects of FOS.
6. Gut homing imprinting is usually ensured by retinoic acid released by CD103+ DCs. As SCFA have been shown to increase the activity of RALDH, the enzyme converting vitamin A into retinoic acid in CD103+ DC (Tan et al, Cell Report 2017), it would be relevant to evaluate the activity of RALDH in CD103+ DC in NC, HFD vs HFD + Fos mice.
7. Fig S3 A shows a decrease of cDC in HFD with around $5 \cdot 10^3$ while in B. HFD are close to $10 \cdot 10^4$. Why are these numbers so different for the same treatment group?
8. In the manuscript cDC2 are either defined as CD103+ or CD103- please correct.
9. FOS affect Th17 in a cDC2 dependent manner but Treg in a cDC2 independent manner. What could be the mechanisms?
10. The fact that FOS improved glucose metabolism partially via IL-17 is very interesting. What could be the mechanisms involved? It would be interesting to supplement HFD fed mice with SFB and determine the impact on glucose metabolism. Similarly does a model of infection with Citrobacter known to increase gut Th17 improve mouse glucose metabolism?

Reviewer #2 (Remarks to the Author):

In this study, Gelineau et al have demonstrated that in the context of high fat diet (HFD), fructose oligosaccharides (FOS) increase the number of intestinal Th17 cells in a cDC2-dependent manner, and the effects of FOS on glucose tolerance are dependent on cDC2s and IL-17.

The following points are raised:

1. In the studies with compositionally defined or purified HFD, the authors use standard rodent chow as the control diet (CD). The use of a general mouse diet (standard chow) is highly discouraged by the scientific community for any dietary intervention study due to various confounding factors including the complexity of grain-based chow and the variation among different chows (PMID: 29371873). Therefore, a purified control diet matched to the HFD should be used.
2. It is unclear whether some of the major findings related to Tregs and Th17 could be driven by FOS even in absence of HFD. Dietary fiber-derived short-chain fatty acids are known to directly promote Tregs. An important control group would be CD+ FOS which is missing in the entire manuscript which could clarify which effect(s) of FOS is dependent of HFD.
3. Although some alteration of microbiota strains has been shown to be associated with the FOS-

mediated changes in Treg and Th17, no experiments were done to confirm the requirement of these strains for the immune phenotypes.

4. Similarly, while gut-homing markers such as CCR9 was suggested to be involved in the effects of FOS, their requirement was not studied.

5. It is unclear how intestinal Th17 affects glucose tolerance. IL-17 has been shown to promote adiposity and glucose intolerance (PMIDs: 21037091, 33859430) while the authors find that the FOS-mediated improvement of glucose tolerance requires IL-17. This contradiction should be addressed and discussed for clarification.

Minor comments:

1. FOS can directly increase colon length (<https://doi.org/10.1101/2022.02.28.482306>) and cecum weight even in absence of high fat, therefore the interpretation that these are the beneficial effects of FOS (lines 162-166) in the context of HFD should be revised.

2. Certain purified diets may cause reduced glucose tolerance compared to chow even in absence of high fat (<https://doi.org/10.1101/2022.02.28.482306>). Therefore, it is unclear whether the difference in glucose tolerance in Fig 1E could be attributed to the fat content of the HFD.

3. The authors should clarify whether the Fig 1G and the '0' time point of figure 1E are the same data from same experiment.

Reviewer #3 (Remarks to the Author):

The manuscript prepared by Gélinau et al. uncovered the role of cDC2 dendritic cells in mediating immune and glucose homeostasis in response to HFD. Further, they further provided a very interesting perspective of the well-supported notion that inclusion or exclusion of specific dietary fibres may be more important than labelling a diet "high-fat" for example. Their evidence supports that a "poor-diet" relates more to the exclusion of dietary fibres (FOS in this study), than the inclusion of fats within the diet. They showed that the impact of FOS on intestinal immunity, gut microbiota, and glucose homeostasis was in part mediated by cDC2 dendritic cells and IL-17 pathways. This manuscript supports a growing body of evidence supporting the positive power of specific dietary fibres in health and disease; a topic of great global interest. My comments to help improve the clarity of the manuscript are included below.

Major comments:

1. Recognizing the importance of differentiating the effects of specific dietary fibres including FOS (ref: PMID: 36183751; PMID: 36323778), it would be best if the authors refer to the specific dietary fibres they tested in their animal model (FOS) throughout, rather than discussing results as a combined "fibre diet" or the impact of "fibre" as they did not measure all fibre types, only FOS. Different fibres display significantly different effects on the host and gut microbiota.

2. Throughout the manuscript it is important to indicate precise scientific results, including indicating if the measurement was the mean/median difference uncovered and p-values. For example: "Already at this time point, HFD increased body weight (Fig. 1A; X%, p=X), weight gain (Fig. 1B; X%, p=X), epididymal (Fig. 1C; X%, p=X), and whole-body fat mass (Fig. 1D; X%, p=X)."

3. Figures should display each animal measured as an individual point on graphs, rather than as a simple mean/median box to accurately display the heterogeneity among animals; the authors only do this in some cases but should do in all cases. Text should also clearly indicate the n-values for animals included in these experiments.

Specific Comments:

1. Line 104 – Do the authors mean to refer to the epididymal fat mass? Please clarify this sentence by accurately describing the measured result, rather than just "epididymal".

2. Line 104 – Fig 1D should include mention of fat mass, "but not lean mass", otherwise that graph

- should be removed if the authors do not intend to make reference to the results in the manuscript.
3. Line 106 – refers to “short-term” without indicating the precise time points; the figures would suggest these measurements in Fig 1E (and perhaps other figures) were not taken at 4-weeks as the prior sentences suggest.
 4. Figure 1E does not display any legible difference between the two lines – both are simply black dots on a line. Please be certain to use a different shape to display HFD or significantly increase the size of the circles. See Fig2F where there is a colour difference between the two.
 5. Figure 1E – can the authors explain why the HFD animals already displayed higher glucose levels even at time 0? Is there a flaw in the model/randomization? Also, why are there no significance stars on the glucose graph? Was there no significant finding here? Same for Fig 2F and remaining curves for glucose throughout the manuscript. If not significant, this should be clear in the text.
 6. Could you please expand the text discussing Figure 1E/2F as there is no discussion of what the glucose levels and AUC results mean; only a very brief mention that “glucose intolerance was noticed”. This is only necessary at first mention and is fine to summarize thereafter.
 7. Fig 1G – only indicates “glucose” on the graph while the text suggests it is a “fasted glucose” level.
 8. Line 108 –please use accurate times rather than “short-term” when discussing results.
 9. Figure 1I/J/K should include representative images of the altered colons of the animals as the length does not seem significant based on the graph displayed (looks to be around 5.8-6.2cm average between the two groups with minor significance). The lack of displaying the individual animals as their own dot points on these graphs makes it difficult to assess accuracy of results.
 10. Fig 1M – do the authors mean to say there was no uncovered significance? That is not clear in the text, yet would seem odd as there is a ~200% increase in the colon. If this was significant, where are the significance stars and reference to the p-value and % increase in the text? Fig 1N also missing p-values or mention that results were not significant.
 11. Fig 1K, Fig. 2M and Fig. S1A inaccurately indicate “caecum” instead of “caecum weight”. Is there a reason that colon length was not included at the later time-point? Was there no significance at this time-point? That would suggest that colon length is not impacted as the earlier results are weak. This would be worth discussing as weight was significantly impacted but not length; interesting.
 12. Line 134: “The dysbiosis reported for Ob/Ob mice likely explains why colonic Th17 cells are already impaired under CD.” This comment should be left for the discussion as the authors did not themselves examine the gut microbiota and are thus, entirely unable to suggest that there was an observed difference in the gut microbiota of the animals within this experiment. While it could be a factor, this is something that would need to be measured within these animals to be mentioned in the results section.
 13. Line 135 – “Overall, these observations suggest that the diet, rather than body weight gain per se, has the strongest impact on ROR γ t+ CD4 T cells homeostasis.” This is not true based on the authors findings that CD v HFD also did not impact the total Treg. Furthermore, the most significant impact ($p=****$) was uncovered between lean CD and Ob/Ob CD rather than diet (CD vs HFD; $p=**$) suggesting that in fact, the body mass does have the most significant impact on ROR γ t+ CD4 T cells in the colon (in contrast to the small intestine). Why were weight of animals not included for lean vs ob/ob animals in Fig S1 to demonstrate there was a significant difference between animal backgrounds and diets?
 14. Fig 2I –Again, only described as “glucose” when this is “plasma glucose”.
 15. Fig 2N – important to note that there was no change in small intestine Treg in Fig 1 and remains no significant change. It is important to discuss these clear differences between the small and large intestine, rather than attempting to force results where there are aren’t any. Why might fat/fibre impact colon over small intestine for total Tregs? What implications do the different immune cell populations between these organ systems support here? This discussion should be expanded. Also, many outliers and significant heterogeneity in Fig 2P which is interesting and worth discussing/highlighting better.
 16. Fig 2Q – were there any significant changes uncovered at the phyla level? Could the percentages +/- and p-values be described?
 17. Fig 2V-X should display full microbe names.
 18. Starting at line 188 – first mention of many microbe species and none are written in full text.
 19. Line 193 – what does “not relevantly regulated” mean? These were not significantly altered? For example, *B. thetaiotamicron* is a fibre-fermenting microbiota and producer of the fermentation

intermediate, succinate. This is certainly relevant.

20. Line 210 – “This is consistent with the drop in SFB abundance we observed upon HFD feeding (Fig. 2S).” This should read “This was associated with” and a figure displaying this association in each animal should be displayed. As the authors did not reintroduce SFB into the animals to demonstrate that SFB causes a shift in T-cell populations in these animals in this study, other correlations or speculations cannot be directly discussed here.

REVIEWER COMMENTS

We thank the reviewers for their comments. All the changes we made in the manuscript were underlined.

Reviewer #1 (Remarks to the Author):

In this manuscript, Gelineau et al investigated the impact of high fat diet and dietary fibre on small intestine and colonic Th17 and Treg and glucose homeostasis. They show that dietary fibre could rescue the effect of high fat diet on gut immunity and glucose homeostasis and that these effects involved type 2 dendritic cells. The manuscript is potentially interesting but too many mechanisms remain elusive. It would strengthen the manuscript if the following were addressed:

1. The absolute number of immune cells should be shown not just the proportion. It would also be good to gate the Th17 and Treg out of CD45+ cells not just CD4+ population.

We thank the reviewer for his/her comment and agree that those pieces of information would add substantial precision to our manuscript. As suggested, we now provide in supplemental figure 1 (S1A-C) the cell numbers as well as the percentage out of CD45+ cells. Of note, absolute numbers are less generally depicted in studies on the intestinal tract as it is sometimes challenging to limit sample to sample variation. Here, it shows that our populations of interest are also decreased in cell numbers and as the percentage of CD45+ cells after 4 weeks of HFD feeding. We also added panels in figure 2 (2O-P) to show the increase in ROR γ t+ T cells subsets numbers after FOS supplementation. Finally, panels were also added in supplemental figure 2 (S2A-B) to display the increase as a percentage of CD45+ cells. Altogether, absolute counts are in agreement with the original data that we presented as a percentage of CD4 T cells.

2. What is the proportion of CD4+ T cells in HFD mice in the small intestine and colon as well as in the thymus to exclude a defect in thymopoiesis.

As suggested by the reviewer, we looked at the proportion of CD4 T cells. There is a decrease in the proportion of CD4 T cells upon HFD in the colon and small intestine (supplemental figure S1B). However, we observe no change in thymus weight (supplemental figure S1I) nor overall cellularity (supplemental figure S1J). Total Tregs and thymic T cell subsets (CD4+ CD8+, CD4+ CD8- and CD4- CD8+) are not modified either (supplemental figure S1K-L). Hence, our intestinal phenotype appears independent of impairments in T cell development in the thymus.

3. What is the proportion of ROR γ t- Treg in the colon and small intestine of HFD fed mice as well as in other sites such as the spleen and the proportion of thymic derived Helios+ Treg.

As suggested by the reviewer, we now provide the proportions and numbers of Tregs subsets in the colon and small intestine of HFD-fed mice (supplemental figure S1D and S1E). We also looked to the spleen and did not observe any alteration at the population level (supplemental figure S1M-Q).

4. What are the blood and gut levels of short chain fatty acids in the HFD mice vs NC and HFD + FOS treated mice? This would confirm the higher gut microbial fermentation activity in FOS treated animals.

It is well established that FOS get fermented into short chain fatty acids (acetate, butyrate, propionate), succinate and lactate in the context of a dietary supplementation (De Vadder F et al. Cell 2014, De Vadder F. et al. Cell Metab 2016, Everard A et al. ISME 2014). Given that we did not previously store samples to analyze short chain fatty acids using gas chromatography/mass spectrometry technology, we have to consider the important literature on the matter as a sufficient grounding. Noteworthy, we confirmed that bacterial fermentation of FOS is elevated as evidenced by the dramatic increase in caecum size (not shown) and weight (Fig 2M) observed in FOS-supplemented mice.

5. FOS has beneficial effects on metabolic and immune outputs potentially through the reshaping of the gut microbiota and/or via the microbial release of SCFA. To understand the mechanism the following should be addressed: 1) does the supplementation of drinking water with SCFA mimic the effects of FOS or 2) does the colonisation of GF mice with HFD microbiota vs HFD+FOS microbiota mimic the effects of FOS.

As suggested by the reviewer, we performed an experiment to determine whether fermentation of FOS into SCFA can recapitulate the effects of fibers on ROR γ t+ CD4 T cells in the mesenteric lymph nodes. We supplemented HFD-fed mice with SCFA (provided in the drinking water) for 4 weeks and show that this treatment corrects the alterations observed in Th17 cells but not ROR γ t+ pTregs (Figure 3I-M). Hence, we conclude that bacterial fermentation of FOS into SCFAs might concur to the increase in Th17 cells, but not of ROR γ t+ Tregs, upon fibers supplementation.

6. Gut homing imprinting is usually ensured by retinoic acid released by CD103+ DCs. As SCFA have been shown to increase the activity of RALDH, the enzyme converting vitamin A into retinoic acid in CD103+ DC (Tan et al, Cell Report 2017), it would be relevant to evaluate the activity of RALDH in CD103+ DC in NC, HFD vs HFD + FOS mice.

CD4 T cells imprinting for gut-homing occurs in the mesenteric lymph nodes (MLNs) and is dependent upon ALDH activity in cDCs and subsequent production of retinoic acid from retinol (Iwata M et al. Immunity 2004, Mora JR et al.

Science 2006). We evaluated ALDH activity in cDC1 and cDC2 in the MLNs of HFD and FOS-supplemented mice. We find that ALDH activity in cDC subsets is increased in HFD-fed mice as compared to controls (supplemental figure S2I). In addition, the fiber supplementation decreases ALDH activity in both cDC populations (supplemental figure S2J). We believe that this regulation could represent a compensatory mechanism in response to the lack of ALDH substrate in HFD-fed animals (while the ALDH assay provides excess substrate to reveal the full potential of ALDH activity). Upon HFD feeding, ALDH expression could be induced to cope with decreased retinol bioavailability. By contrast, FOS supplementation would ameliorate retinol bioavailability in a way that favors cDCs' ability to efficiently imprint CD4 T cells for gut-homing. Thus, a possible hypothesis is that HFD feeding decreases the bioavailability of retinol to dendritic cells in the intestine, impairing the generation and gut-homing imprinting of Th17 cells. Indeed, while the CD and the HFD have similar vitamin A content, published data revealed that tissue retinol levels are decreased upon HFD feeding (Trasino SE et al. Sci Rep 2015). Previous work revealed that retinol-derived retinoic acid activates epithelial retinoic acid receptors (RARs), thus inducing the expression of its target genes such as SAAs (Gattu S et al. PNAS 2019). Importantly, SAA induction is also critical during SFB colonization to prime Th17 (Ivanov II et al. Cell 2009) and has also been shown to transport retinol to myeloid cells and support adaptive immunity (Bang Y et al. Science 2021). This axis might be impaired by HFD and restored by fibers. Indeed, we showed that the SFB is decreased in HFD-fed animals compared to CD animals and re-established by fibers (figure 2S). We also observed that the HFD significantly decreases the expression of *Saa2*, but not *Saa1*, whereas the FOS supplementation increases the expression of both (data not shown), suggesting impaired retinoic acid signaling upon HFD and its amelioration by FOS. Moreover, it has been recently highlighted that the gut microbiota, namely SFB, synthesizes retinoic acid from dietary vitamin A (Woo V et al. Cell Host Microbes 2021). As SFB is decreased upon HFD feeding and elevated upon FOS supplementation, SFB-derived retinoic acid might also play a role here. Altogether, the literature and our data hint towards retinoids bioavailability and retinoic acid-induced signals impairments upon HFD feeding, which would be corrected by dietary fibers.

7. Fig S3 A shows a decrease of cDC in HFD with around $5 \cdot 10^3$ while in B. HFD are close to $10 \cdot 10^4$. Why are these numbers so different for the same treatment group?

We thank the reviewer for pointing out our mistake. To calculate absolute numbers, we use beads that we include in our tubes at a defined number (10000 beads per tube). We check our calculation and realized that the number of beads was not properly reported in the excel file related to this figure (10 times less beads than what was actually added), leading to an overestimation of the actual number of cells. We corrected this mistake.

8. In the manuscript cDC2 are either defined as CD103+ or CD103- please correct.

We thank the reviewer for this comment. Although the status of CD11b⁺ CD103⁻ cDC is controversial, we chose to consider that cDC2 encompass both CD11b⁺ CD103⁺ and CD11b⁺ CD103⁻ cDC as suggested in previous studies (Scott CL et al. Mucosal Immunol 2015, Cerovic V et al. Mucosal Immunol 2013). We acknowledge that the role and maturity state of the latter is less well defined. Nonetheless, it was shown that CD11b⁺ CD103⁻ cDC can induce Th17 cells (Scott CL et al. Mucosal Immunol 2015, Cerovic V et al. Mucosal Immunol 2013). Importantly, in CD11c-cre x *Irf4*^{flox/flox} mice, while both subsets are altered, the CD11b⁺ CD103⁺ cDC2 subset completely disappears while CD103⁻ CD11b⁺ are reduced but not absent. However, we cannot exclude them from our interpretation, which is why we conclude that the whole cDC2 compartment is responsible for our results.

9. FOS affect Th17 in a cDC2 dependent manner but Treg in a cDC2 independent manner. What could be the mechanisms?

We thank the reviewer for this very interesting question. pTregs have been shown to be induced by both cDC1 and cDC2 in models of oral tolerance, and by recently discovered ROR γ t⁺ APCs in oral tolerance models and in the steady state (Esterhazy D et al. Nat Immunol 2016, Persson E et al. Immunity 2013, Akagbosu B et al. Nature 2022, Kedmi R et al. Nature 2022). Current consideration is that the pTregs-inducing antigen-presenting cells (APCs) depend upon the nature (microbial or dietary) and doses of antigens, as well as the timeline of antigen presentation (Russler-Germain E et al. ELife 2021). Hence, oral tolerance seems to develop as a result of redundant mechanisms. Our data suggests the existence of compensatory mechanisms for ROR γ t⁺ pTregs generation by other APCs in the absence of cDC2. Alternatively, it could be that the environmental changes induced by HFD and fibers that modulate ROR γ t⁺ Tregs are not sampled by cDC2 in the first place, in contrast to the ones from the chow diet (as pTregs are decreased at the steady state in CD11c-cre x *Irf4*^{flox/flox} mice). Importantly, as we focused on population size and not TCR sequences, maintenance of the pool does not imply qualitative similarity in terms of TCR specificities. Along the same line, whether the expression of ROR γ t in Tregs is dependent on dendritic cells is unknown, and it might be that pTregs are altered in the absence of cDC2 in a way that cannot be assessed using the current markers.

Interestingly, we note that the redundancy of pTregs-inducing mechanisms is not sufficient to maintain the pool of pTregs upon HFD. This could also support our hypothesis of environmental changes rather than a cDC intrinsic impairment.

In any case, deeper study of the mechanisms that maintain pTregs in the absence of cDC2 in FOS-supplemented mice exceeds the scope of this manuscript, as our effects are ascribed to Th17.

10. The fact that FOS improved glucose metabolism partially via IL-17 is very interesting. What could be the mechanisms involved? It would be interesting to supplement HFD fed mice with SFB and determine the impact on glucose metabolism. Similarly does a model of infection with *Citrobacter* known to increase gut Th17 improve mouse glucose metabolism?

We thank the reviewer for his/her interest in our phenotype. We have several hints regarding what mechanisms are involved. IL-17 is a very important cytokine for intestinal homeostasis, and we now show that the cDC2-Th17 axis mediates the beneficial impact of dietary fibers on the expression of genes involved in the maintenance of metabolic health: the hormone PYY, the lectins RegIIIg and RegIIIb, and the mucins Muc2 and Muc3 (supplemental figure S6). PYY is known as a regulator of insulin sensitivity, while lectins and mucins are key players in both symbiosis and intestinal immunoregulation. Hence, we think the fiber-cDC2-Th17 axis concurs to maintain intestinal physiology through several routes that build up towards benefits at the scale of the organism.

Regarding Th17-inducing bacterial strains and their impact on immunity and metabolism, recent work from Kawano Y and al. (Cell 2022) very nicely demonstrated that supplementing SFB during a 4-weeks HFD feeding period (same as ours) ameliorated glucose homeostasis (Kawano Y et al. Cell 2022). We added a sentence acknowledging this important result in the discussion.

Regarding *Citrobacter*-induced inflammatory Th17 cells, it is important to note that they differ from SFB-induced homeostatic Th17 cells as they mostly secrete IFN γ and IL-22 (Omenetti S et al. Immunity 2019). We have data indicating that HFD decreases IL17-secreting cells (data not shown). Hence, increasing the *Citrobacter*-induced subset would not help drawing a conclusion in our case.

Reviewer #2 (Remarks to the Author):

In this study, Gelineau et al have demonstrated that in the context of high fat diet (HFD), fructose oligosaccharides (FOS) increase the number of intestinal Th17 cells in a cDC2-dependent manner, and the effects of FOS on glucose tolerance are dependent on cDC2s and IL-17.

The following points are raised:

1. In the studies with compositionally defined or purified HFD, the authors use standard rodent chow as the control diet (CD). The use of a general mouse diet (standard chow) is highly discouraged by the scientific community for any dietary intervention study due to various confounding factors including the complexity of grain-based chow and the variation among different chows (PMID: 29371873). Therefore, a purified control diet matched to the HFD should be used.

We thank the reviewer for this very important precision and fully agree with the importance of using matched diets as controls for high-fat purified diet. We think that our manuscript supports this progress, since our main argued point focuses on the fibers, which is added in drinking water on top of the HFD, creating a perfectly matched control. Even though our first observation compares a purified and a non-purified diet, our subsequent choice of method allows us to study one precise part of the diet in equally purified intakes. We think our manuscript fits in the conversation on matching diets, as we make the point that that fiber content is extremely important to match for both immunological and metabolic studies.

2. It is unclear whether some of the major findings related to Tregs and Th17 could be driven by FOS even in absence of HFD. Dietary fiber-derived short-chain fatty acids are known to directly promote Tregs. An important control group would be CD+ FOS which is missing in the entire manuscript which could clarify which effect(s) of FOS is dependent of HFD.

The regular chow diet contains a high load of dietary fibers, and we indeed didn't assess the effect of FOS supplementation on chow-fed animals in the first place. Instead, we asked whether FOS were able to improve immune and metabolic parameters in the context of HFD (which is deprived of dietary fibers) feeding.

To address the question raised by the reviewer, we treated chow diet-fed animals with FOS and assess Th17 cells and ROR γ t⁺ pTregs homeostasis in the mesenteric lymph nodes. We observe that overall Tregs as well as ROR γ t⁺ Tregs are not impacted by the supplementation of fibers on top of the CD (supplemental figure S2E). However, Th17 cells display a tendency to be increased (supplemental figure S2F), and Th17 cells that express CCR9 are increased (supplemental figure S2F). Thus, FOS supplementation marginally increases Th17 cells, but not ROR γ t⁺ Tregs, in CD-fed animals. This new set of results is included in the updated manuscript. This suggests that the fiber content of the chow diet we used in our study is sufficient to fully support intestinal ROR γ t⁺ development and maintenance.

3. Although some alteration of microbiota strains has been shown to be associated with the FOS-mediated changes in Treg and Th17, no experiments were done to confirm the requirement of these strains for the immune phenotypes.

We thank the reviewer for this suggestion and agree that establishing causality between microbiota changes and immune phenotype would increase the strength of our manuscript.

However, recent work from Kawano Y and al. (Cell 2022) coopts our results on SFB reduction upon HFD feeding as they show that supplementing HFD-fed animals (same HFD than in our study) with SFB during 4 weeks increases Th17 cells and ameliorates glucose homeostasis (Kawano Y and al. Cell 2022). We added a sentence highlighting those results in our discussion.

Concerning ROR γ ⁺ pTregs, we consider that this question exceeds the scope of our manuscript since only Th17 are involved in our phenotype.

4. Similarly, while gut-homing markers such as CCR9 was suggested to be involved in the effects of FOS, their requirement was not studied.

We agree with the reviewer that the functional involvement of gut-homing receptors was not experimentally confirmed in our manuscript.

We can argue that the role of gut-homing markers in metabolic and immune regulation in response to the diet has been explored. In a 2015 pioneer study on the importance of the intestinal immune system in overall metabolic health, Luck et al. used Itg β 7 knock-out animals and studied the consequences of diminished gut-homing on metabolic pathology. They show that Itg β 7 KO animals have worsened metabolic pathology in the context of HFD feeding (Luck H et al. Cell Metab 2015).

In our study, we would need an overexpression model in order to maintain gut-homing receptors in the context of HFD feeding to ask whether this would improve the immune and metabolic phenotype. However, such a model does not exist yet.

5. It is unclear how intestinal Th17 affects glucose tolerance. IL-17 has been shown to promote adiposity and glucose intolerance (PMIDs: 21037091, 33859430) while the authors find that the FOS-mediated improvement of glucose tolerance requires IL-17. This contradiction should be addressed and discussed for clarification.

As mentioned by the reviewer, IL-17 was first discovered in the context of autoimmune diseases and can have pro-inflammatory properties. As shown in the cited work, IL-17 and Th17 cells in the liver or adipose tissue, which start to expand after a minimum of 8 weeks of HFD (to be compared to the 4 weeks period of feeding in our study), usually have a detrimental impact on metabolic homeostasis (MacDougall CE et al. Cell Metab 2018, Bertola A et al. Diabetes 2012, Zuniga LA et al. J Immunol 2010, Xu R et al. ABBS 2013, Teijeiro A et al. Nat Metab 2021). By contrast, our results resonate with another line of research arguing that intestinal Th17 cells have beneficial effects on intestinal physiology, mucosal integrity and symbiosis to limit inflammation. For example, IL-17 favors IgA and antimicrobial peptides production, and maintain tight epithelial junctions, thus maintaining the intestinal barrier function (Cao AT et al. J Immunol 2012, Ishigame H et al. Immunity 2011, Lee JS et al. Immunity 2015). In metabolic pathogenesis, other groups' results also point out specifically to intestinal Th17 cells and their cytokines as beneficial rather than detrimental (Wang X et al. Nature 2014, Hong CP et al. Gastroenterology 2017, Garidou L et al. Cell Metab 2015).

Our results show that intestinal Th17 cells are already decreased after 4 weeks of HFD, while adipose tissue and liver Th17 are not modified by the HFD at this time point. In addition, while the fibers prevented the loss of intestinal Th17 cells, it has no impact on adipose tissue and liver Th17 cells. These data are presented in supplemental figure 3. Importantly, it seems that timeline matters as studies showing increase in Th17 cells or IL-17 in the adipose tissue or liver systematically rely on a much longer time on high-fat diet as compared to our current study (superior to 8 weeks in all studies, and superior to 12 weeks in most) (MacDougall CE et al. Cell Metab 2018, Bertola A et al. Diabetes 2012, Chen Y et al. PLoS One 2014, Zuniga LA et al. J Immunol 2010, Xu R. et al ABBS 2013, Teijeiro A et al. Nat Metab 2021). At those longer timepoints, Th17 cells have been proposed to fuel obesity-associated comorbidities. Hence, we believe that IL-17 inhibition in our experimental setting has a dominant effect on intestinal Th17. We believe the restricted timeframe of our study (4 weeks) enables to restrain the effects in other organs, considering that the intestine is the first metabolic organ in contact with the dietary content. This difference in intestine versus adipose tissue and liver Th17 cells alteration is consistent with literature suggesting that intestinal inflammation is a primary event in dietary-induced inflammation, that precedes adipose tissue and liver inflammation (Luck H et al. Cell Metabolism 2015, Ding S et al. PLoS One 2010, Siracusa F et al. Nat Immunol 2023). Finally, IL-17 inhibition in other organs would be unlikely to worsen the metabolic phenotype as we observed here. This is now presented and discussed in the manuscript (adjacent to the supplemental figure 3 results).

Minor comments:

1. FOS can directly increase colon length (<https://doi.org/10.1101/2022.02.28.482306>) and cecum weight even in absence of high fat, therefore the interpretation that these are the beneficial effects of FOS (lines 162-166) in the context of HFD should be revised.

In our study, we indeed observed that FOS supplementation was able to increase colon length and cecum weight in HFD-fed animals. By contrast, in chow diet-fed mice, FOS supplementation on top of our chow diet, only increased caecum weight but not colon weight and length (data not shown). We added a sentence in this section to specify the impact of FOS supplementation in the context of chow diet feeding.

2. Certain purified diets may cause reduced glucose tolerance compared to chow even in absence of high fat (<https://doi.org/10.1101/2022.02.28.482306>). Therefore, it is unclear whether the difference in glucose tolerance in Fig 1E could be attributed to the fat content of the HFD.

We agree with the reviewer comment. Our point here is to insist that the impact of the HFD on glucose metabolism is at least partially accountable on the absence of fibers, rather than on the sole fat content. Our work ultimately uses one purified diet, the HFD, and fiber supplementation in the drinking water was provided to a group of animals, creating a perfectly matched control with regards to the diet used.

3. The authors should clarify whether the Fig 1G and the '0' time point of figure 1E are the same data from same experiment.

We thank the reviewer for raising this concern. It is indeed the same data from the same experiment and we have clarified this point in the legend of the figure.

Reviewer #3 (Remarks to the Author):

The manuscript prepared by G lineau et al. uncovered the role of cDC2 dendritic cells in mediating immune and glucose homeostasis in response to HFD. Further, they further provided a very interesting perspective of the well-supported notion that inclusion or exclusion of specific dietary fibres may be more important than labelling a diet "high-fat" for example. Their evidence supports that a "poor-diet" relates more to the exclusion of dietary fibres (FOS in this study), than the inclusion of fats within the diet. They showed that the impact of FOS on intestinal immunity, gut microbiota, and glucose homeostasis was in part mediated by cDC2 dendritic cells and IL-17 pathways. This manuscript supports a growing body of evidence supporting the positive power of specific dietary fibres in health and disease; a topic of great global interest. My comments to help improve the clarity of the manuscript are included below.

We thank the reviewer for his/her positive appraisal of our study.

Major comments:

1. Recognizing the importance of differentiating the effects of specific dietary fibres including FOS (ref: PMID: 36183751; PMID: 36323778), it would be best if the authors refer to the specific dietary fibres they tested in their animal model (FOS) throughout, rather than discussing results as a combined "fibre diet" or the impact of "fibre" as they did not measure all fibre types, only FOS. Different fibres display significantly different effects on the host and gut microbiota.

We have chosen to use FOS as a model for soluble and fermentable fibers as they are often used in the metabolic diseases field. However, as pointed out by the reviewer, FOS do not fully represent the complexity and heterogeneity of dietary fibers. This is an important precision and we have adapted our text by replacing occurrences of fibers with FOS where relevant.

2. Throughout the manuscript it is important to indicate precise scientific results, including indicating if the measurement was the mean/median difference uncovered and p-values. For example: "Already at this time point, HFD increased body weight (Fig. 1A; X%, p=X), weight gain (Fig. 1B; X%, p=X), epididymal (Fig. 1C; X%, p=X), and whole-body fat mass (Fig. 1D; X%, p=X)."

We agree with the reviewer that this would add precision to data description. However, this may also make the text harder to read. We checked and it seems that Nature Communications publications usually do not follow this format. This will be discussed with the editorial team to see which format they prefer and changes will be made if necessary.

3. Figures should display each animal measured as an individual point on graphs, rather than as a simple mean/median box to accurately display the heterogeneity among animals; the authors only do this in some cases but should do in all cases. Text should also clearly indicate the n-values for animals included in these experiments.

Varying graphic styles (histograms versus dot plots) was an aesthetic choice, as a paper displaying only one type of graph would be less visually appealing. We believe this is a consideration that is very important to keep the reader's attention. We are very rigorous in systematically displaying mean and standard error on each histogram to account for data distribution. N values are also systematically included in the figure description so that the information is openly available without making the text heavier. Hence, we do not think our statistical approach to be flawed by representation of data. Importantly, for a number of key and important findings we used individual points to display interindividual variability.

Specific Comments:

1. Line 104 – Do the authors mean to refer to the epididymal fat mass? Please clarify this sentence by accurately describing the measured result, rather than just “epididymal”.

Indeed, epididymal referred to “fat mass” (“epididymal and whole-body fat mass”) in order to avoid repetition of “fat mass” in the same sentence. We have adapted the text to increase clarity.

2. Line 104 – Fig 1D should include mention of fat mass, “but not lean mass”, otherwise that graph should be removed if the authors do not intend to make reference to the results in the manuscript.

As pointed out by the reviewer, the description did not match our figure and we have now adapted the text to clarify this point.

3. Line 106 – refers to “short-term” without indicating the precise time points; the figures would suggest these measurements in Fig 1E (and perhaps other figures) were not taken at 4-weeks as the prior sentences suggest.

“Short-term” refers to 4 weeks of diet as it is commonly accepted as a short-term feeding period in contrast to long term (more than 1 month). As requested by the reviewer we added the precision within the text to avoid confusion. We reduced the occurrence of “short term” and added the time length in parenthesis when necessary.

4. Figure 1E does not display any legible difference between the two lines – both are simply black dots on a line. Please be certain to use a different shape to display HFD or significantly increase the size of the circles. See Fig2F where there is a colour difference between the two.

We have increased the size of the dots for better clarity.

5. Figure 1E – can the authors explain why the HFD animals already displayed higher glucose levels even at time 0? Is there a flaw in the model/randomization? Also, why are there no significance stars on the glucose graph? Was there no significant finding here? Same for Fig 2F and remaining curves for glucose throughout the manuscript. If not significant, this should be clear in the text.

Increased fasting glucose levels at baseline are a well-known feature of HFD feeding. This validated that HFD feeding efficiently altered glucose metabolism in our cohort as expected.

Regarding the statistical analysis of glucose tolerance test, significance for the whole graph is made by comparing the area under the curve (AUC) obtained for each animal. This is a consensual way to quantify difference between glucose tolerance test curves that accounts for the difference between the overall curves and not between particular x points. This is why it is represented attached to the curves but not directly on them. The purpose of each AUC histograms is to quantify difference between the curves displayed right before. Of course, statistical differences could also be observed on the glucose tolerance curves but comparing the AUCs is a more correct way to compare the animals' efficiency to clear glucose from the circulation.

6. Could you please expand the text discussing Figure 1E/2F as there is no discussion of what the glucose levels and AUC results mean; only a very brief mention that “glucose intolerance was noticed”. This is only necessary at first mention and is fine to summarize thereafter.

We thank the reviewer for its comment and we adapted our text for a better clarity.

7. Fig 1G – only indicates “glucose” on the graph while the text suggests it is a “fasted glucose” level.

Indeed, the figure was unclear. We added the precision in the figure description.

8. Line 108 –please use accurate times rather than “short-term” when discussing results.

We mitigated our use of “short term” and added “4 weeks” as a precision to avoid all confusion.

9. Figure 1I/J/K should include representative images of the altered colons of the animals as the length does not seem significant based on the graph displayed (looks to be around 5.8-6.2cm average between the two groups with minor significance). The lack of displaying the individual animals as their own dot points on these graphs makes it difficult to assess accuracy of results.

The decrease is modest but it is statistically significant and we now show it as dot points as suggested by the reviewer for increased clarity.

10. Fig 1M – do the authors mean to say there was no uncovered significance? That is not clear in the text, yet would seem odd as there is a ~200% increase in the colon. If this was significant, where are the significance stars and reference to the p-value and % increase in the text? Fig 1N also missing p-values or mention that results were not significant.

We thank the reviewer for his comment. We added information regarding the statistical significance on both panels.

11. Fig 1K, Fig. 2M and Fig. S1A inaccurately indicate “caecum” instead of “caecum weight”. Is there a reason that colon length was not included at the later time-point? Was there no significance at this time-point? That would suggest

that colon length is not impacted as the earlier results are weak. This would be worth discussing as weight was significantly impacted but not length; interesting.

As suggested by the reviewer, we indicated “caecum weight” instead of “caecum”.

Unfortunately, we did not measure colon length in the later time-point. However, we think that colon length is likely to be decreased at this time as shown in another study (Chassaing B et al. Am J Physiol Gastrointest Liver Physiol 2015).

12. Line 134: “The dysbiosis reported for Ob/Ob mice likely explains why colonic Th17 cells are already impaired under CD.” This comment should be left for the discussion as the authors did not themselves examine the gut microbiota and are thus, entirely unable to suggest that there was an observed difference in the gut microbiota of the animals within this experiment. While it could be a factor, this is something that would need to be measured within these animals to be mentioned in the results section.

We understand the point raised by the reviewer and changed the phrasing to acknowledge that we did not directly address it.

13. Line 135 – “Overall, these observations suggest that the diet, rather than body weight gain per se, has the strongest impact on ROR γ t+ CD4 T cells homeostasis.” This is not true based on the authors findings that CD v HFD also did not impact the total Treg. Furthermore, the most significant impact ($p=****$) was uncovered between lean CD and Ob/Ob CD rather than diet (CD vs HFD; $p=**$) suggesting that in fact, the body mass does have the most significant impact on ROR γ t+ CD4 T cells in the colon (in contrast to the small intestine). Why were weight of animals not included for lean vs ob/ob animals in Fig S1 to demonstrate there was a significant difference between animal backgrounds and diets?

We thank the reviewer for his comments and we agree that this might be more complicated. Regarding the body weight, it has been shown that feeding Ob/Ob mice a HFD increases weight gain in comparison with low-fat diet fed Ob/Ob mice (Mercer SW et al. J Nutr 1987, Wang JH et al. Front Microbiol 2019). Here, we also observed that HFD feeding increased body weight in Ob/Ob mice (Lean CD: 23.4 ± 0.6 ; Ob/Ob CD: 41.4 ± 2.1 ; Ob/Ob HFD: 64.6 ± 0.9).

To follow the reviewer’s suggestion, we have changed the text to mention that both the diet and body weight impact on ROR γ t+ CD4 T cells.

14. Fig 2I –Again, only described as “glucose” when this is “plasma glucose”.

We thank the reviewer and have adapted the figure.

15. Fig 2N – important to note that there was no change in small intestine Treg in Fig 1 and remains no significant change. It is important to discuss these clear differences between the small and large intestine, rather than attempting to force results where there are aren’t any. Why might fat/fibre impact colon over small intestine for total Tregs? What implications do the different immune cell populations between these organ systems support here? This discussion should be expanded. Also, many outliers and significant heterogeneity in Fig 2P which is interesting and worth discussing/highlighting better.

We agree with the reviewer that small intestinal Tregs do not change upon HFD and HFD+FOS, and this is now clearly stated in the manuscript. We now mention that “In HFD-fed animals, the frequency of total Tregs was unaltered in the small intestine but slightly diminished in the colon”, and that “FOS did not significantly impact total Tregs in the small intestine, while it increased them in the colon”.

In contrast with the small intestine that receives a mixture of digestible and undigestible elements, the colon receives complex carbohydrates and proteins that escaped digestion in the small intestine. Those complex nutrients can provide substrate to the microbiota, leading to higher fermentation. Hence, both the food antigen and the bacterial antigen landscape is radically different between the colon and the small intestine. Functional differences between small intestinal and colonic Tregs are yet to be elucidated but there is no doubt that they evolve in different antigenic landscapes.

We agree that there was some heterogeneity in the data presented in figure 2P, but statistical tests did not identify outliers.

16. Fig 2Q – were there any significant changes uncovered at the phyla level? Could the percentages +/- and p-values be described?

To follow the reviewer’s suggestion, we added a supplemental Table (Figure S2D) to better highlight the significant changes.

17. Fig 2V-X should display full microbe names.

We changed the names as suggested by the reviewer.

18. Starting at line 188 – first mention of many microbe species and none are written in full text.

This indeed escaped our attention and we adapted our manuscript to include full species names.

19. Line 193 – what does “not relevantly regulated” mean? These were not significantly altered? For example, B.

thetaitomicron is a fibre-fermenting microbiota and producer of the fermentation intermediate, succinate. This is certainly relevant.

We agree that our phrasing was not accurate. "Not relevantly regulated" meant that the variations of the microbe did not follow the ROR γ t⁺ Tregs regulation pattern (i.e not diminished by HFD and increased by FOS). We have now changed the text.

20. Line 210 – "This is consistent with the drop in SFB abundance we observed upon HFD feeding (Fig. 2S)." This should read "This was associated with" and a figure displaying this association in each animal should be displayed. As the authors did not reintroduce SFB into the animals to demonstrate that SFB causes a shift in T-cell populations in these animals in this study, other correlations or speculations cannot be directly discussed here. We changed the text according the reviewer's suggestion. Unfortunately, we couldn't display the association for each animal as we do not have SFB levels for every single animal.

We thank the reviewer for their in-depth comments that allow us to achieve a higher precision in our manuscript, and we hope to have succeeded in addressing their concerns.

REVIEWER COMMENTS

Reviewer #1 (Remarks to the Author):

All my comments have been addressed. In my opinion, the manuscript is now suitable for publication in Nature Communications.

Reviewer #2 (Remarks to the Author):

No further comments

Reviewer #3 (Remarks to the Author):

Thank you to the authors for providing edits. Some of the prior key comments were disregarded and still need to be dealt with. It is unclear what specific changes were made as the authors simply indicate "changes were made" without indicating the line or text that was altered.

While the text still ambiguously suggests that the authors evaluated a complex fibre solution, they did not evaluate a "high fibre diet" but instead they examined the impact of the single B-fructan fibre, fructooligosaccharide. This needs to be reflected in the title and throughout the manuscript. For example even in the discussion the authors claim "fibres" in the drinking water impacted results when only consumption of one single fibre, FOS was examined. In fact, it cannot be concluded that FOS directly resulted in these effects, but rather "consumption of FOS" as SCFA and other byproducts could have induced the described results. There is no natural diet where FOS would exclusively be consumed as total dietary fibre, thus it is essential to highlight the effects of this study are related to FOS and not fibres. A study including at least one other unrelated dietary fibre (not another B-fructan but still a fermentable fibre for example) would need to be performed to more broadly suggest the findings relate to "fibres". This is particularly related to the fact that all of these impacts could be, and likely are, related to production of SCFA from FOS and not merely the consumption of any fibre (for example PMID: 34233192); introducing SCFA orally would not recapitulate the impacts of SCFA produced by natural fermentation in the hind-gut as oral SCFA would be absorbed inappropriately rather than engaging with colonic gut cells.

I respectfully request again that dot plots be included. It is confusing and concerning that the authors feel it appropriate to only include dot plots sometimes and are unwilling to effectively display the heterogeneity of their findings in other graphs. Fig 2P heterogeneity is one reason why it is essential to show dot plots for human and animal studies. Particularly as the field continues to demonstrate that there are "responders" and "non-responders" within these nutritional and microbial study populations, providing clear evidence of how these animal responses clustered (tight vs spread out) is paramount. Please correct this.

Line 205 - SFB was already defined in the intro - it is not required again here.

REVIEWER COMMENTS

We thank the reviewers for their comments. All the changes we made in the manuscript were written in blue and underlined (previous changes were still underlined in black).

Reviewer #1 (Remarks to the Author):

All my comments have been addressed. In my opinion, the manuscript is now suitable for publication in Nature Communications.

We thank the reviewer for his/her positive appraisal of our study.

Reviewer #2 (Remarks to the Author):

No further comments.

We thank the reviewer for his/her positive appraisal of our study.

Reviewer #3 (Remarks to the Author):

Thank you to the authors for providing edits. Some of the prior key comments were disregarded and still need to be dealt with. It is unclear what specific changes were made as the authors simply indicate "changes were made" without indicating the line or text that was altered.

We thank the reviewer for his/her comments

During the 1st round of revision, we have underlined in black the changes we made in the text.

The new changes we operated were written in blue color and underlined.

We have copy-pasted (see below) the previous round of revision and now indicated the line(s) or text that was altered when necessary (this is indicated in orange color).

Sorry for not providing this previously.

Round 1:

“ Reviewer #3 (Remarks to the Author):

The manuscript prepared by G lineau et al. uncovered the role of cDC2 dendritic cells in mediating immune and glucose homeostasis in response to HFD. Further, they further provided a very interesting perspective of the well-supported notion that inclusion or exclusion of specific dietary fibres may be more important than labelling a diet “high-fat” for example. Their evidence supports that a “poor-diet” relates more to the exclusion of dietary fibres (FOS in this study), than the inclusion of fats within the diet. They showed that the impact of FOS on intestinal immunity, gut microbiota, and glucose homeostasis was in part mediated by cDC2 dendritic cells and IL-17 pathways. This manuscript supports a growing body of evidence supporting the positive power of specific dietary fibres in health and disease; a topic of great global interest. My comments to help improve the clarity of the manuscript are included below.

We thank the reviewer for his/her positive appraisal of our study.

Major comments:

1. Recognizing the importance of differentiating the effects of specific dietary fibres including FOS (ref: PMID: 36183751; PMID: 36323778), it would be best if the authors refer to the specific dietary fibres they tested in their animal model (FOS) throughout, rather than discussing results as a combined “fibre diet” or the impact of “fibre” as they did not measure all fibre types, only FOS. Different fibres display significantly different effects on the host and gut microbiota.

We have chosen to use FOS as a model for soluble and fermentable fibers as they are often used in the metabolic diseases field. However, as pointed out by the reviewer, FOS do not fully represent the complexity and heterogeneity of dietary fibers. This is an important precision and we have adapted our text by replacing occurrences of fibers with FOS where relevant.

-> This has been done throughout the manuscript. All text changes were underlined in black

2. Throughout the manuscript it is important to indicate precise scientific results, including indicating if the measurement was the mean/median difference uncovered and p-values. For example: “Already at this time point, HFD increased body weight (Fig. 1A; X%, p=X), weight gain (Fig. 1B; X%, p=X), epididymal (Fig. 1C; X%, p=X), and whole-body fat mass (Fig. 1D; X%, p=X).”

We agree with the reviewer that this would add precision to data description. However, this may also make the text harder to read. We checked and it seems that Nature Communications publications usually do not follow this format. This will be discussed with the editorial team to see which format they prefer and changes will be made if necessary.

3. Figures should display each animal measured as an individual point on graphs, rather than as a simple mean/median box to accurately display the heterogeneity among animals; the authors only do this in some cases but should do in all cases. Text should also clearly indicate the n-values for animals included in these experiments.

Varying graphic styles (histograms versus dot plots) was an aesthetic choice, as a paper displaying only one type of graph would be less visually appealing. We believe this is a consideration that is very important to keep the reader's attention. We are very rigorous in systematically displaying mean and standard error on each histogram to account for data distribution. N values are also systematically included in the figure description so that the information is openly available without making the text heavier. Hence, we do not think our statistical approach to be flawed by representation of data. Importantly, for a number of key and important findings we used individual points to display interindividual variability.

-> This has been done throughout the manuscript. All text changes were underlined in black

Specific Comments:

1. Line 104 – Do the authors mean to refer to the epididymal fat mass? Please clarify this sentence by accurately describing the measured result, rather than just “epididymal”.

Indeed, epididymal referred to “fat mass” (“epididymal and whole-body fat mass”) in order to avoid repetition of “fat mass” in the same sentence. We have adapted the text to increase clarity.

-> This change can be found in line 103

2. Line 104 – Fig 1D should include mention of fat mass, “but not lean mass”, otherwise that graph should be removed if the authors do not intend to make reference to the results in the manuscript.

As pointed out by the reviewer, the description did not match our figure and we have now adapted the text to clarify this point.

-> This change can be found in line 104

3. Line 106 – refers to “short-term” without indicating the precise time points; the figures would suggest these measurements in Fig 1E (and perhaps other figures) were not taken at 4-weeks as the prior sentences suggest.

“Short-term” refers to 4 weeks of diet as it is commonly accepted as a short-term feeding period in contrast to long term (more than 1 month). As requested by the reviewer we added the precision within the text to avoid confusion. We reduced the occurrence of “short term” and added the time length in parenthesis when necessary.

-> This change can be found in line 106

4. Figure 1E does not display any legible difference between the two lines – both are simply black dots on a line. Please be certain to use a different shape to display HFD or significantly increase the size of the circles. See Fig2F where there is a colour difference between the two.

We have increased the size of the dots for better clarity.

-> This change can be found in Fig 1E

5. Figure 1E – can the authors explain why the HFD animals already displayed higher glucose levels even at time 0? Is there a flaw in the model/randomization? Also, why are there no significance stars on the glucose graph? Was there no significant finding here? Same for Fig 2F and remaining curves for glucose throughout the manuscript. If not significant, this should be clear in the text.

Increased fasting glucose levels at baseline are a well-known feature of HFD feeding. This validated that HFD feeding efficiently altered glucose metabolism in our cohort as expected.

Regarding the statistical analysis of glucose tolerance test, significance for the whole graph is made by comparing the area under the curve (AUC) obtained for each animal. This is a consensual way to quantify difference between glucose tolerance test curves that accounts for the difference between the overall curves and not between particular x points. This is why it is represented attached to the curves but not directly on them. The purpose of each AUC histograms is to quantify difference between the curves displayed right before. Of course, statistical differences could also be observed on the glucose tolerance curves but comparing the AUCs is a more correct way to compare the animals' efficiency to clear glucose from the circulation.

6. Could you please expand the text discussing Figure 1E/2F as there is no discussion of what the glucose levels and AUC results mean; only a very brief mention that “glucose intolerance was noticed”. This is only necessary at first mention and is fine to summarize thereafter.

We thank the reviewer for its comment and we adapted our text for a better clarity.

-> This change can be found in lines 104-107

7. Fig 1G – only indicates “glucose” on the graph while the text suggests it is a “fasted glucose” level. Indeed, the figure was unclear. We added the precision in the figure description.

-> This change can be found in the figure legends (line 805 and 825).

8. Line 108 –please use accurate times rather than “short-term” when discussing results.

We mitigated our use of “short term” and added “4 weeks” as a precision to avoid all confusion.

-> This has been changed throughout the manuscript.

9. Figure 1I/J/K should include representative images of the altered colons of the animals as the length does not seem significant based on the graph displayed (looks to be around 5.8-6.2cm average between the two groups with minor significance). The lack of displaying the individual animals as their own dot points on these graphs makes it difficult to assess accuracy of results.

The decrease is modest but it is statistically significant and we now show it as dot points as suggested by the reviewer for increased clarity.

-> Fig 1J, 1K and 1L have been modified for increased clarity.

10. Fig 1M – do the authors mean to say there was no uncovered significance? That is not clear in the text, yet would seem odd as there is a ~200% increase in the colon. If this was significant, where are the significance stars and reference to the p-value and % increase in the text? Fig 1N also missing p-values or mention that results were not significant.

We thank the reviewer for his comment. We added information regarding the statistical significance on both panels.

-> Fig 1M and 1N have been modified to add statistical significance.

11. Fig 1K, Fig. 2M and Fig. S1A inaccurately indicate “caecum” instead of “caecum weight”. Is there a reason that colon length was not included at the later time-point? Was there no significance at this time-point? That would suggest that colon length is not impacted as the earlier results are weak. This would be worth discussing as weight was significantly impacted but not length; interesting.

As suggested by the reviewer, we indicated “caecum weight” instead of “caecum”.

Unfortunately, we did not measure colon length in the later time-point. However, we think that colon length is likely to be decreased at this time as shown in another study (Chassaing B et al. Am J Physiol Gastrointest Liver Physiol 2015).

-> Fig 1K, 2M and S1F have been modified to replace “caecum” by “caecum weight”.

12. Line 134: “The dysbiosis reported for Ob/Ob mice likely explains why colonic Th17 cells are already impaired under CD.” This comment should be left for the discussion as the authors did not themselves examine the gut microbiota and are thus, entirely unable to suggest that there was an observed difference in the gut microbiota of the animals within this experiment. While it could be a factor, this is something that would need to be measured within these animals to be mentioned in the results section.

We understand the point raised by the reviewer and changed the phrasing to acknowledge that we did not directly address it.

-> This change can be found in line 151.

13. Line 135 – “Overall, these observations suggest that the diet, rather than body weight gain per se, has the strongest impact on ROR γ t+ CD4 T cells homeostasis.” This is not true based on the authors findings that CD v HFD also did not impact the total Treg. Furthermore, the most significant impact (p=****) was uncovered between lean CD and Ob/Ob CD rather than diet (CD vs HFD; p=**) suggesting that in fact, the body mass does have the most significant impact on ROR γ t+ CD4 T cells in the colon (in contrast to the small intestine). Why were weight of animals not included for lean vs ob/ob animals in Fig S1 to demonstrate there was a significant difference between animal backgrounds and diets?

We thank the reviewer for his comments and we agree that this might be more complicated. Regarding the body weight, it has been shown that feeding Ob/Ob mice a HFD increases weight gain in comparison with low-fat diet fed Ob/Ob mice (Mercer SW et al. J Nutr 1987, Wang JH et al. Front Microbiol 2019). Here, we also observed that HFD feeding increased body weight in Ob/Ob mice (Lean CD: 23.4 \pm 0.6; Ob/Ob CD: 41.4 \pm 2.1; Ob/Ob HFD: 64.6 \pm 0.9).

To follow the reviewer’s suggestion, we have changed the text to mention that both the diet and body weight impact on ROR γ t+ CD4 T cells.

-> This change can be found in line 153.

14. Fig 2I –Again, only described as “glucose” when this is “plasma glucose”.

We thank the reviewer and have adapted the figure.

-> Fig 1E, 1G, 2F, 2I and 6F have been modified accordingly.

15. Fig 2N – important to note that there was no change in small intestine Treg in Fig 1 and remains no

significant change. It is important to discuss these clear differences between the small and large intestine, rather than attempting to force results where there are aren't any. Why might fat/fibre impact colon over small intestine for total Tregs? What implications do the different immune cell populations between these organ systems support here? This discussion should be expanded. Also, many outliers and significant heterogeneity in Fig 2P which is interesting and worth discussing/highlighting better.

We agree with the reviewer that small intestinal Tregs do not change upon HFD and HFD+FOS, and this is now clearly stated in the manuscript. We now mention that "In HFD-fed animals, the frequency of total Tregs was unaltered in the small intestine but slightly diminished in the colon", and that "FOS did not significantly impact total Tregs in the small intestine, while it increased them in the colon".

-> These changes can be found in line 120-121 and 181-182.

In contrast with the small intestine that receives a mixture of digestible and undigestible elements, the colon receives complex carbohydrates and proteins that escaped digestion in the small intestine. Those complex nutrients can provide substrate to the microbiota, leading to higher fermentation. Hence, both the food antigen and the bacterial antigen landscape is radically different between the colon and the small intestine. Functional differences between small intestinal and colonic Tregs are yet to be elucidated but there is no doubt that they evolve in different antigenic landscapes.

We agree that there was some heterogeneity in the data presented in figure 2P, but statistical tests did not identify outliers.

16. Fig 2Q – were there any significant changes uncovered at the phyla level? Could the percentages +/- and p-values be described?

To follow the reviewer's suggestion, we added a supplemental Table (Figure S2D) to better highlight the significant changes.

-> A table was added in Fig S2D to complement the information provided in Fig 2Q..

17. Fig 2V-X should display full microbe names.

We changed the names as suggested by the reviewer.

-> Fig 2V-X have been modified accordingly.

18. Starting at line 188 – first mention of many microbe species and none are written in full text.

This indeed escaped our attention and we adapted our manuscript to include full species names.

-> This has been changed throughout the manuscript (from line 204 to 221).

19. Line 193 – what does "not relevantly regulated" mean? These were not significantly altered? For example, *B. thetaiotamicron* is a fibre-fermenting microbiota and producer of the fermentation intermediate, succinate. This is certainly relevant.

We agree that our phrasing was not accurate. "Not relevantly regulated" meant that the variations of the microbe did not follow the $ROR\gamma^+$ Tregs regulation pattern (i.e not diminished by HFD and increased by FOS). We have now changed the text.

-> This has been changed in lines 216-220.

20. Line 210 – "This is consistent with the drop in SFB abundance we observed upon HFD feeding (Fig. 2S)." This should read "This was associated with" and a figure displaying this association in each animal should be displayed. As the authors did not reintroduce SFB into the animals to demonstrate that SFB causes a shift in T-cell populations in these animals in this study, other correlations or speculations cannot be directly discussed here.

We changed the text according to the reviewer's suggestion. Unfortunately, we couldn't display the association for each animal as we do not have SFB levels for every single animal.

-> This has been in line 236.

While the text still ambiguously suggests that the authors evaluated a complex fibre solution, they did not evaluate a "high fibre diet" but instead they examined the impact of the single B-fructan fibre, fructooligosaccharide. This needs to be reflected in the title and throughout the manuscript. For example even in the discussion the authors claim "fibres" in the drinking water impacted results when only consumption of one single fibre, FOS was examined. In fact, it cannot be concluded that FOS directly resulted in these effects, but rather "consumption of FOS" as SCFA and other byproducts could have induced the described results. There is no natural diet where FOS would exclusively be consumed as total dietary fibre, thus it is essential to highlight the effects of this study are related to FOS and not fibres. A study including at least one other unrelated dietary fibre (not another B-fructan but still a fermentable fibre for example) would need to be performed to more broadly suggest the findings relate to "fibres". This is particularly related to the fact that all of these impacts could be, and likely are, related to production of SCFA from FOS and not merely the consumption of any fibre (for example PMID: 34233192); introducing SCFA orally would not recapitulate the impacts of SCFA produced by natural fermentation in the hind-gut as oral SCFA would be absorbed inappropriately rather than engaging with colonic gut cells.

We thank the reviewer for his comment and we have made the changes accordingly.

The title has now been changed.

We have also replaced fibers by fructooligosaccharides (or FOS) throughout the manuscript. The new changes appear in blue and underlined throughout the manuscript. Our previous changes in that sense still appear underlined in black throughout the manuscript.

In addition, we now mention "FOS supplementation", "FOS administration" or "FOS intake" (and not FOS) throughout the manuscript. The new changes appear in blue and underlined throughout the manuscript while our previous changes still appear underlined in black throughout the manuscript.

I respectfully request again that dot plots be included. It is confusing and concerning that the authors feel it appropriate to only include dot plots sometimes and are unwilling to effectively display the heterogeneity of their findings in other graphs. Fig 2P heterogeneity is one reason why it is essential to show dot plots for human and animal studies. Particularly as the field continues to demonstrate that there are "responders" and "non-responders" within these nutritional and microbial study populations, providing clear evidence of how these animal responses clustered (tight vs spread out) is paramount. Please correct this.

We have made the changes accordingly to the reviewer suggestion. Dot pots were used to depict our results in the main and supplemental figures.

Line 205 - SFB was already defined in the intro - it is not required again here.

This has been modified (line 203)

We thank the reviewer for his/her positive appraisal of our study.

REVIEWERS' COMMENTS

Reviewer #3 (Remarks to the Author):

Sufficient changes have been made.